# Dissecting the Role of Positional Encoding in Length Generalization

## Abstract

Length generalization (LG) is a persistent challenge for Transformers. Despite recent studies improving the models' LG capability, its underlying mechanisms are still underexplored. To better understand LG, we propose that LG requires alignment of the model's inductive bias with the task's computational structure, and validate this view with experiments on Transformers. Focusing on iterative tasks (e.g., Polynomial Iteration, Parity, Binary Copy), we systematically analyze different Positional Encodings (PEs) and find that the misalignment persists for Transformers: the structural bias of softmax attention and computational biases from PEs destabilize LG under extrapolation. Notably, Transformers without positional encoding (NoPE) could show partial LG capability, potentially because implicit position encoding through hidden-state statistics and contextual token distributions preserves the consistent computation in extrapolation, though these signals decay with length, leaving the encoding misaligned with the task. Building on this mechanistic analysis, we introduce a lightweight enhancement—value-side relative coding with logit rescaling—that better aligns inductive bias with task structure. This sustains iterative computation and improves LG, offering insights for future PE design.

## 1 Introduction

Transformers are prevalent in numerous fields, and their length generalization (LG) capability has recently led to extensive discussion (Anil et al., 2022; Zhao et al., 2023a;b). Length generalization refers to the model's ability to extrapolate from training sequences of bounded length to longer test sequences. To understand LG, Zhou et al. (2023); Abbe et al. (2024); Xiao & Liu (2025) investigate why transformers generalize and how LG might be achieved by exploring their expressive power. Other studies include directly improving LG by modifying or designing new Transformer-based algorithms (Golovneva et al., 2024; Munkhdalai et al., 2024), or by introducing innovations in Positional Encodings (PEs) (Peng et al., 2023; Li et al., 2023; Hua et al., 2024).

PEs are widely explored in the context of LG (Gu et al., 2025; Dufter et al., 2022). In particular, Kazemnejad et al. (2023) evaluated different PEs on synthetic reasoning tasks, showing that models even with No Positional Encoding (NoPE) can exhibit better LG than other PEs, and confirming that NoPE can encoding positional information, though the origin of this ability remains unclear. At the same time, a subset of synthetic reasoning tasks—especially mathematical ones—remain hard for nearly all PEs (Zhou et al., 2023; Lee et al., 2023; Gr'egoire Del'etang et al., 2022). While specially designed PEs sometimes improve LG on such tasks (Shen et al., 2023; Lee et al., 2023; Zhou et al., 2024), they are often task-specific and non-robust rather than general solutions.

To better understand LG, we propose that it requires an *alignment* between the task's computational structure and the model's inductive bias—its inherent preference for certain ways of computation (Xiao & Liu, 2024; 2025). This inspires us to consider a special class of synthetic reasoning problems—iterative tasks such as Polynomial Iteration, Parity, and Binary Copy—which naturally decompose into step-by-step computational updates (Cabannes et al., 2024; Wei et al., 2022; Chung et al., 2024). Transformers trained on these tasks exhibit a distinctive attention pattern that aligns with the stepwise computation structure of the tasks (Cabannes et al., 2024), hence suitable for us to analyze the relation between the inductive bias and task structures as well as the LG behavior. Building on this setting, we conduct a systematic study of how positional encodings (PEs) in Transformers

influenced LG. We compare APE (Vaswani et al., 2017), T5 (Raffel et al., 2020), ALiBi (Press et al., 2021), YaRN (Peng et al., 2023), FIRE (Li et al., 2023), RoPE (Su et al., 2024), and NoPE across the three iterative tasks, with the goal of uncovering how PEs shape the model's inductive bias and whether this bias aligns with the task's structure required for LG.

**Our contributions are:**

- We establish that LG relies on the alignment between the model's inductive bias and the task's computational structure. We visualize the attention map to probe the internal computation flow of Transformers, finding that, while Transformers exhibit partial alignment with iterative tasks, misalignments still remain: the structural bias of softmax attention and the computational biases from PEs may destabilize LG.

- We investigate the effects of PEs on LG and find variance and limited performances across all encodings. For common PEs such as RoPE, APE, and NoPE, we show performance degradation indeed stems from the misalignment of PEs and attention with the requirements of iterative tasks. Notably, NoPE achieves the best LG performance, possibly because, in specific settings, its hidden-state statistics (mean and variance) and contextual token distributions preserve positional consistency. Yet this signal fades with length, leaving its alignment with the task incomplete.

- Building on these insights, we propose a lightweight enhancement—value-side relative coding with logit rescaling—that better aligns model inductive bias with task structure. This sustains the internal recurrence computation and improves LG, offering guidance for future PE design.

## 2 PRELIMINARY

**Attention Mechanism** We briefly review causal self-attention in Transformers (Vaswani et al., 2017; Radford et al., 2019). Consider an input sequence $X = (\mathbf{x}_1, \ldots, \mathbf{x}_L)$, where each $\mathbf{x}_n \in \mathbb{R}^d$ is a $d$-dimensional embedding and $n$ indexes token position ($1 \leq n \leq L$). Each $\mathbf{x}_n$ is linearly mapped to queries, keys, and values: $\mathbf{q}_n = W_Q \mathbf{x}_n$, $\mathbf{k}_n = W_K \mathbf{x}_n$, $\mathbf{v}_n = W_V \mathbf{x}_n$, with $W_Q, W_K, W_V \in \mathbb{R}^{d \times d}$. Under the causal mask, the output at position $n$ attends only to tokens $1{:}n$, with weights

$$\alpha_{n,i} = \frac{\exp(\langle \mathbf{q}_n, \mathbf{k}_i \rangle / \sqrt{d})}{\sum_{i'=1}^{n} \exp(\langle \mathbf{q}_n, \mathbf{k}_{i'} \rangle / \sqrt{d})}, \quad 1 \leq i \leq n, \tag{1}$$

and the contextualized representation $\mathbf{z}_n = \sum_{i=1}^{n} \alpha_{n,i} \mathbf{v}_i \in \mathbb{R}^d$. This formulation highlights that each output $\mathbf{z}_n$ is a weighted sum of past values, with weights determined by the similarity between the current query and preceding keys under the causal mask.

**Positional Encodings** *Absolute positional encodings (APE)* assign each position a unique vector, either via fixed sinusoidal functions (Vaswani et al., 2017) or by learning embeddings during training (Brown et al., 2020). In this paper we refer to fixed sinusoidal functions (Vaswani et al., 2017) as APE. *Relative positional encodings (RPE)* encode pairwise distances, which is particularly effective for long contexts. Representative examples include T5 (Raffel et al., 2020), ALiBi (Press et al., 2021), and RoPE (Su et al., 2024), as well as extensions like YaRN (Peng et al., 2023) and FIRE (Li et al., 2023), most of which inject information via the QK path to bias attention logits. In addition, Transformers can also operate without explicit PEs, *NoPE*, relying solely on causal masking (Haviv et al., 2022; Kazemnejad et al., 2023), which enforces autoregressive order and allows the model to learn positional relations implicitly from sequence tokens.

**Distance–Attention Bias in RPEs.** Most RPEs inject distance information into the $QK$ path, so that the logit between query $n$ and key $i$ takes the form $\ell_{ni} = \langle \mathbf{q}_n, T_\delta \mathbf{k}_i \rangle + g(\delta)$, $\delta = n - i$. Here $T_\delta$ and $g(\delta)$ depend only on the relative offset, which means attention strength is explicitly coupled to distance. Depending on the design, this coupling can attenuate long-range attention (e.g., ALiBi (Press et al., 2021), RoPE (Su, 2021; Su et al., 2024)) or take more flexible forms (e.g., FIRE (Li et al., 2023)), but in all cases distance systematically biases attention patterns. We refer to this general phenomenon as a *distance-attention bias*.

**Iterative Tasks.** We define an *iterative task* as a sequence-to-output problem whose solution can be decomposed into repeated local updates. Let the input be $X = (x_1, \ldots, x_L) \in \mathcal{X}^L$, the corresponding states $S = (s_1, \ldots, s_L) \in \mathcal{S}^L$, and the final output $y \in \mathcal{Y}$, where $\mathcal{X}$ is the input

space, $\mathcal{S}$ the state space, and $\mathcal{Y}$ the output space. The overall mapping is $F : \mathcal{X}^L \to \mathcal{Y}$ with $F(x_1, \ldots, x_L) = y$. An iterative task is characterized by the fact that $F$ can be achieved by a *iterative computational structure* $s_1 = x_1$, $s_t = f(s_{t-1}, x_t)$, $t = 2, \ldots, L$ for some update rule $f : \mathcal{S} \times \mathcal{X} \to \mathcal{S}$. The final output is obtained from the last state $y = g(s_L)$, where $g : \mathcal{S} \to \mathcal{Y}$ is typically the identity ($y = s_L$) in many tasks. Take the Polynomial Iteration task as an example, let $x_t \in \mathbb{F}_p$ (the finite field $\mathbb{Z}/p\mathbb{Z}$) and fix a function $f : \mathbb{F}_p \times \mathbb{F}_p \to \mathbb{F}_p$. We set $s_1 = x_1$ and update $s_t = f(s_{t-1}, x_t) \bmod p$ for $t \geq 2$. A common affine instance is $s_t = (s_{t-1} \cdot x_t + 1) \bmod 5$.

**Input Sequence Formats**  Following prior work on sequence modeling (Sutskever et al., 2014; Cabannes et al., 2024), we introduce special tokens to delimit different segments: *BoS* (Beginning of Sequence) marks the start of the input, *EoI* (End of Input) separates the input segment from the reasoning trajectory, and *EoS* (End of Sequence) marks the termination of the entire sequence. Thus, an iterative task is serialized into a sequence that contains both the input $x_{1:L}$ and the state trajectory $s_{1:L}$ leading to the final output. For example, given input `[BoS, 1, 2, 3, 4, EoI]`, the trajectory unfolds as $s_1 = 1$, $s_2 = 3$, $s_3 = 0$, $s_4 = 1 \pmod 5$. The original target is simply `[1, EoS]`, whereas the full trajectory is represented as `[1, 3, 0, 1, EoS]`. As illustrated in Figure 1(a), such serialized sequences follow a strict computational structure: inputs $x_t$ are indexed from BoS with offset $\Delta_1 = t$, while states $s_{t-1}$ are indexed from EoI with offset $\Delta_2 = t-1$, preserving the invariant $\Delta_1 = \Delta_2 + 1$. This ensures that each update retrieves the correct pair $(s_{t-1}, x_t)$ regardless of sequence length.

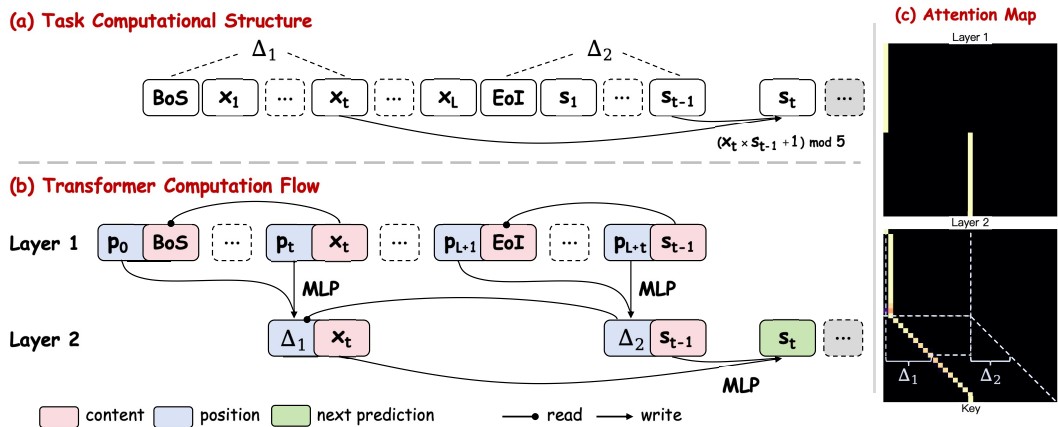

**Figure 1:** **(a)** Computational structure of Polynomial Iteration. **(b)** Computation flow in Transformers with PEs. **(c)** Anchor-based attention in Layer 1 and ladder-like attention in Layer 2. All three figures reflect the similar patterns that select $(s_{t-1}, x_t)$ based on $\Delta_1 = \Delta_2 + 1$ to predict $s_t$.

## 3  LENGTH GENERALIZATION REQUIRES MODEL-TASK ALIGNMENT

In this section, we first explain that the success of LG relies on the alignment between the model's inductive bias and the computational structure of the task.

We consider a model trained on sequences of length $T$ that approximates a function $F(x_1, \ldots, x_T)$. At test time, when sequence length extends beyond training ($T \to T + k$), it must approximate a new function $G(x_1, \ldots, x_T, x_{T+1}, \ldots, x_{T+k})$, defined over a larger input domain, where the trained function $F$ corresponds only to a restricted subspace: $F(x_1, \ldots, x_T) = G(x_1, \ldots, x_T, 0_{1:k})$. Since the optimization during training constrains only this subspace, the remaining regions of $G$'s domain remain random initialization. As a result, predictions on longer sequences lack meaningful structure, explaining the catastrophic degradation observed during length extrapolation (Xiao & Liu, 2025).

However, when we introduce an inductive bias into the model and impose a clear constraint on the task, the extrapolation domain becomes narrower, as the inductive bias restricts the hypothesis space by favoring certain rules over others. When such inductive bias aligns with the task's computational structure, the domain is restricted, making extrapolation feasible. To illustrate this interaction, consider a cumulative multiplication task $y = \prod_{i=1}^{n} x_i$ as an example. We define a multiplicative

model with independent parameters, $\hat{y} = \prod_{i=1}^{n} \beta_i x_i$. During training, the model can fit the task well by learning $\prod_{i=1}^{n} \beta_i$ to be close to 1, yielding correct outputs for $n \leq T$. However, for $n > T$, the unseen parameters $\beta_{T+1}, \beta_{T+2}, \dots$ remain random, leading to failure in LG. From the perspective of the model's inductive bias, if we change from independent parameters to weight sharing ($\beta_i = \beta, \ \forall i \in \{1, \dots, n\}$), the same update rule applies at every step, and extrapolation succeeds once training learns $\beta = 1$. Conversely, from the perspective of the task's computational structure, if the task itself is instead a multiplicative process with position-specific coefficients $y = \prod_{i=1}^{n} \beta_i x_i$, then the model's weight-sharing bias ($\beta_i = \beta$) enforces the wrong rule, leading to failure in extrapolation. This example illustrates that length generalization is not a property of the task or the model alone, but of their alignment.

> **Insight.** Length generalization is not a property of the model or the task in isolation, but emerges only when the model's inductive bias aligns with the task's computational structure.

## 3.1 Inductive Bias of Transformer Aligns with Iterative Tasks

LG is feasible when the inductive bias of models aligns with the computational structure of tasks. For iterative tasks, each update depends only on $(x_t, s_{t-1})$, with $x_t$ located relative to BoS and EoI ($\Delta_1 = \Delta_2 + 1$). Thus, solving requires extracting key tokens (BoS, EoI, $s_{t-1}$, $x_t$) and reapplying the update rule $f(x_t, s_{t-1})$. Thus a model that can generalize in such tasks should filter tokens from contexts and sustain iterative computation beyond training lengths.

Transformers potentially satisfy this requirement: attention supports token selection, and parameter sharing enforces reuse across steps. Based on this intuition, we hypothesize a plausible computation flow that Transformers may use to solve iterative tasks, summarized as a two-step procedure (Figure 1 (b)). *Layer 1: Relative-position extraction via content-indexed attention.* The query of $s_{t-1}$ attends to the key of EoI, while the query of $x_t$ attends to the key of BoS. These content-based lookups pass positional information into the hidden states of $s_{t-1}$ and $x_t$, producing the relative offsets $\Delta_2 = t-1$ and $\Delta_1 = t$. *Layer 2: Content routing via position-indexed attention.* In the subsequent attention, the query of $s_{t-1}$ matches the key of $x_t$ by enforcing the relation $\Delta_1 = \Delta_2 + 1$. This unique query–key match routes the content of $x_t$ into $s_{t-1}$, after which the MLP combines the two streams to yield the updated state $s_t$.

We validate this flow by inspecting attention maps during training under diverse PEs. On most PEs, we observe consistent attention patterns. As shown in Figure 1(c), patterns reflect the computation flow in Figure 1(b): layer 1 exhibits anchor-based attention on BoS and EoI, indicating that models use anchors to calculate positional offsets ($\Delta_1, \Delta_2$). Layer 2 shows a ladder-like pattern, where each reasoning token focuses on its corresponding input token, aligning with the routes that $s_{t-1}$ selects $x_t$ via the relation $\Delta_1 = \Delta_2 + 1$ to form $f(s_{t-1}, x_t)$. Even RPEs like RoPE, though imposing distance–attention bias (Section 2), still focus on key tokens during training (Appendix D.2). Prior work (Cabannes et al., 2024) also noted similar attention patterns, termed Iteration Head, as a sign of learning iterative computation, but did not extend to other PEs or link inner flows to task structure.

Taken together, the pattern in Figure 1 (c) reflects the internal computation flow in (b), showing how Transformer solves tasks in (a): Transformer can simulate token filtering and reapply the update rule required by the task's structure. Thus, Transformer inductive bias may potentially align with the task, and attention patterns offer a useful lens to assess whether such computation remains stable.

## 3.2 Fragility of Alignment in Length Extrapolation

Though the inductive bias of Transformers exhibit potential alignment with iterative task structure, this alignment is fragile in extrapolation. In Transformer's architecture, two key sources of bias may disturb its stability for LG. First, softmax attention distributes weights across the full context, so longer sequences add interference weakening focus on key tokens. Second, PEs may impose computational biases—e.g., RoPE enforces distance–attention bias (Su, 2021)—preventing consistent attention regardless of distance. Together, these biases misalign with the task's structure. Training may enforce attention patterns in Figure 1 (c), but under extrapolation the biases may interfere with iterative computation, leaving LG unreliable. Since such misalignments remain only a potential concern, we next need to conduct experiments to examine whether such biases indeed affect LG in extrapolation.

**Insights.** Transformers can simulate iterative computation. Yet such alignment is fragile: the structural bias of softmax attention and the computational biases of PEs may disrupt stable anchor-based routing as context grows. The key challenge for LG is not whether Transformers can learn iterative computation, but whether their inductive biases allow this process to remain robust under extrapolation.

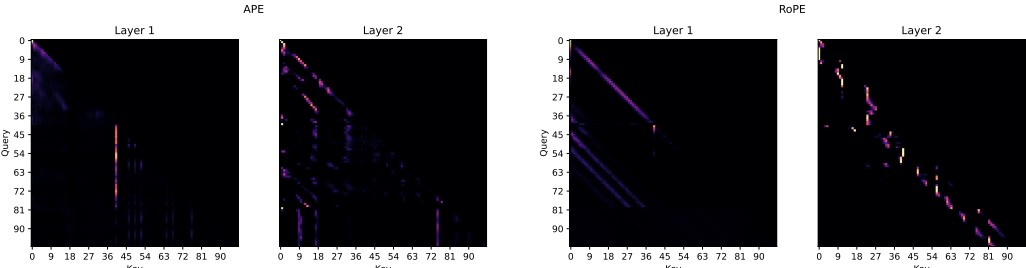

**Figure 2:** Illustration of attention patterns degradation on Polynomial Iteration under o.o.d. lengths. The model is a 2-layer Transformer trained on input lengths (problem lengths) 1–16. The attention map is shown for a test sample with input length 39 (total sequence length 81), where the attention pattern breaks down.

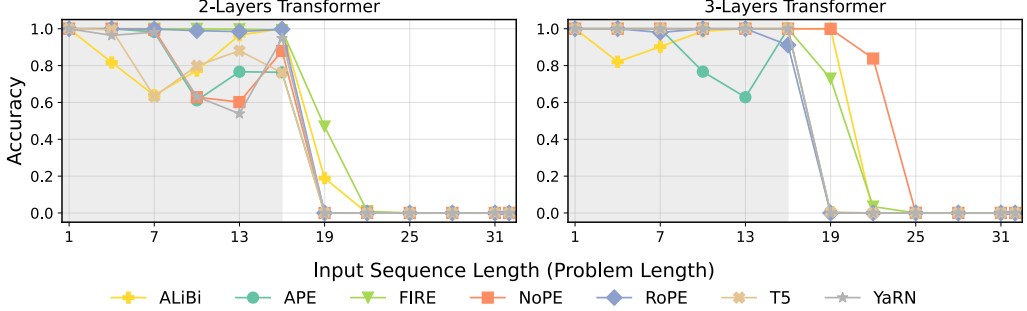

**Figure 3:** Train and test (input-length beyond 16) accuracy on Polynomial Iteration of all PEs as input sequence length increases. Left: performances of 2-layer Transformers. Right: performances of 3-layer Transformers.

## 4 IMPACT OF PEs ON LENGTH EXTRAPOLATION

### 4.1 EXPERIMENTAL OBSERVATIONS

**Overall.** We evaluate both the attention pattern and task accuracy of different PEs under out of distribution (o.o.d.) sequence lengths (see Appendix B for experimental details). From Figure 2, As sequences grow longer than training, the attention pattern collapses across nearly all encodings. This breakdown is mirrored in accuracy: across all PEs, performance drops sharply from training to testing, confirming that none achieve true LG (Figure 3).

**Common PEs.** APE performs poorly, consistent with its known weakness in extrapolation, while RoPE degrades even faster despite its reputation for stronger LG. In contrast, NoPE shows a shift: the 2-layer model is hard to fit training, but the 3-layer achieves the best accuracy among all PEs, though it still declines rapidly under extrapolation. To probe further, we examine step-level accuracy of reasoning tokens (the accuracy of certain subproblem) that remain within the total sequence length of training but appear under longer input length (problem length). This setup isolates how enlarged context windows and different PEs affect subproblem predictions. Results (Figure 4) show RoPE collapsing almost immediately, APE following abruptly, and NoPE degrading more gradually, retaining resistance longer though still ultimately affected.

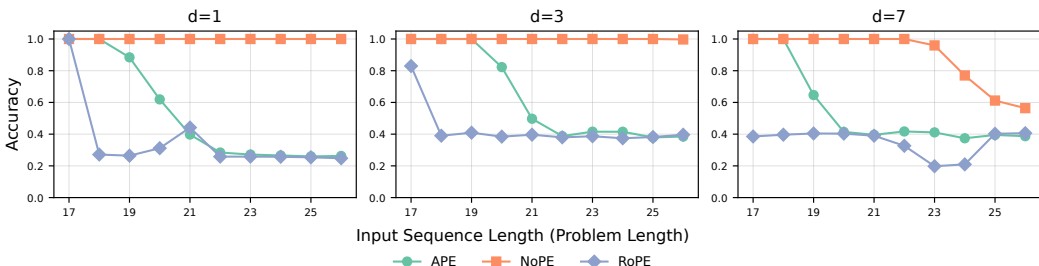

**Figure 4:** Test accuracy of a 3-layer Transformer on Polynomial Iteration. We increase input length beyond the training range and evaluate $s_{t-1}$'s next-step token accuracy while keeping it certain distance to EoI (d=1, 3, 7) and making its absolute position less than the training maximum (Appendix A.3). Each subfigure represents different distance $s_{t-1}$ to EoI.

### 4.2 WHY ATTENTION COLLAPSES?

Our experiments show that even small changes incured by o.o.d. lengths collapse attention patterns and iterative computation formed in training. To better understand this phenomenon, we analyze RoPE as a case study, showing how its distance-attention bias affects computation.

RoPE modifies the attention logit between query $n$ and key $i$ as $\ell_{ni} = \langle \mathbf{q}_n, R(\theta(n-i)) \mathbf{k}_i \rangle$, where $R(\theta(n-i))$ is a rotation operator with angle proportional to the relative distance $\delta = n-i$. This directly couples similarity to $\delta$: as $|n-i|$ grows, the rotation introduces oscillations and attenuations, making long-range attention weaker (Su, 2021). However, solving the iterative tasks requires two forms of distance-agnostic focus. First, $s_{t-1}$ must attend to EoI and $x_t$ to BoS in anchor extraction which only depend on the anchor's content regardless of lengths and contexts. Second, $s_{t-1}$ must select $x_t$ based on $\Delta_1 = \Delta_2 + 1$, a fixed offset relation invariant to context size. The pair $(s_{t-1}, x_t)$ should always be strongly linked no matter how far apart they are. Taken together, *RoPE conflicts with this requirement by injecting distance-dependent rotation into QK similarity, forcing attention strength to vary with $n-i$.* As a result, the anchor focus in Layer 1 and the offset-based match in Layer 2 degrade as distances grow, causing the collapse of the computation formed during training when extrapolating. Combined with the case analysis of the different performance of PEs, we suggest that PEs could introduce their own computational bias affecting task alignment.

Besides PEs' bias, we suggest that softmax attention also contributes to collapse. Figure 4 shows that even without distance-attention bias, APE and NoPE degrade as input length increases. Since Figure 4 isolates accuracy on a fixed subtask while only extending the context, the decline indicates that information introduced by irrelevant tokens in longer contexts may disrupt computation.

> **Insight.** The reason why Transformers fail to maintain attention patterns and achieve LG on iterative tasks: the model's inductive biases—computational (from PEs) and structural (from softmax attention)—are misaligned with the inherent computational structure of the task.

## 5 MECHANISM OF NOPE: HOW IMPLICIT ENCODING WORKS

As shown in Figure 3 and 4, NoPE attains the best LG performance, however, it is not exempt from degradation: its accuracy still declines under longer contexts. This motivates a deeper analysis of how NoPE encodes positional information and why it only supports partial LG.

### 5.1 ENCODING POSITIONS VIA HIDDEN-STATE STATISTICS

Prior studies have suggested that NoPE can implicitly encode positional information. Kazemnejad et al. (2023) showed that the first layer of NoPE captures absolute positions, while the second layer captures relative positions, providing a constructive proof of its potential to encode position. In parallel, Chi et al. (2023) argued that positional signals emerge through the variance of hidden states, and Su (2024) further demonstrated in a simplified setting that the variance of a $d$-dimensional hidden-state vector encodes the absolute position $n$ as $\sigma^2/n$.

Building on these insights, we further explore the encoding capability of NoPE based on coordinate-wise statistics of hidden states. Since in most Transformer tasks a BoS token is prepended, we also consider its role in encoding positions.

**Proposition 1** (Statistical Encoding under NoPE). *Consider one layer attention with uniform weights $\alpha_{ni} = \frac{1}{n}$ for $i \leq n$, with BoS token prepended. The attention output at position $n$ is*

$$\mathbf{z}_n = \frac{\mathbf{v}_{\text{BoS}} + \sum_{i=1}^{n-1} \mathbf{v}_i}{n}, \tag{2}$$

*where $\mathbf{v}_{\text{BoS}} = (b_1, \ldots, b_d)$ and $\mathbf{v}_i = (v_{i,1}, \ldots, v_{i,d})$.*

*Define the coordinate-wise mean, variance, and adjacent difference as:*

$$\bar{z}_n = \tfrac{1}{d} \sum_{j=1}^{d} z_{n,j}, \qquad \widehat{\text{Var}}(z_n) = \tfrac{1}{d} \sum_{j=1}^{d} (z_{n,j} - \bar{z}_n)^2, \qquad \Delta(z_{n+1}, z_n) = \tfrac{1}{d} \sum_{j=1}^{d} (z_{n+1,j} - z_{n,j}). \tag{3}$$

*We assume $b_j \sim \mathcal{N}(\mu_2, \sigma_2^2)$ and $v_{i,j} \sim \mathcal{N}(\mu, \sigma^2)$, coordinates are i.i.d. across $d$ dimension. And for large $d$ the empirical averages concentrate by the law of large numbers. Then the following scaling laws hold:*

$$\bar{z}_n \approx \mu + \frac{\mu_2 - \mu}{n}, \qquad \widehat{\text{Var}}(z_n) \approx \frac{\sigma_2^2 + (n-1)\sigma^2}{n^2}, \qquad \Delta(z_{n+1}, z_n) \approx -\frac{\mu_2 - \mu}{n(n+1)}. \tag{4}$$

**Interpretation.** Proof of Proposition 1 is deferred to Appendix C.1. Proposition 1 shows that NoPE encodes positions through simple statistics of $\mathbf{z}_n$: the mean decays as $O(1/n)$ from $\mu_2$ toward $\mu$, variance scales as $O(1/n)$, and adjacent differences vanish at $O(1/n^2)$. These yield a positional signal that is monotonic, bounded, and decaying. Intuitively, the BoS token serves as an anchor: position $n$ is represented by the residual BoS contribution in the hidden state. This explains why NoPE provides positional cues at moderate lengths.

## 5.2 Contextual Token Distributions in Sequences

To better explain how NoPE exploits statistical differences (Eq. 4) between tokens to encode positional information, we propose the following perspective: the root cause of statistical differences lies in the contextual token distribution of the original sequence itself. This view provides a more intuitive and transparent explanation, while also offering a unified criterion for determining which types of input sequences can, or cannot, carry positional information.

**Proposition 2** (Contextual Token Distributions in Original Sequences). *Consider one layer attention. Each token $c_i$ belongs to one of $C$ categories with embedding $\mathbf{v}_{c_i} \in \mathbb{R}^d$, where we absorb the shared value projection $W_v$ into the embeddings so $\mathbf{v}_c$ denotes the value representation. Let $V = [\mathbf{v}_1, \ldots, \mathbf{v}_C] \in \mathbb{R}^{d \times C}$ collect category embeddings, For position $n$, denote attention weights by $\alpha_{ni}$ with $i \leq n$ and $\sum_{i \leq n} \alpha_{ni} = 1$. Define contextual token distributions over the prefix $1{:}n$ by*

$$I_c(n) = \{ i \leq n : c_i = c \}, \qquad \beta_{c,n} := \sum_{i \in I_c(n)} \alpha_{ni}, \qquad \boldsymbol{\beta}_n = (\beta_{1,n}, \ldots, \beta_{C,n}) \in \Delta^{C-1}. \tag{5}$$

*Then the attention output is a linear embedding of this distribution:*

$$\mathbf{z}_n = V \boldsymbol{\beta}_n. \tag{6}$$

*If $\alpha_{ni} = \frac{1}{n}$ for all $i \leq n$, then $\beta_{c,n} = \frac{|I_c(n)|}{n}$, i.e., $\boldsymbol{\beta}_n$ equals the category proportions in the prefix.*

**Interpretation.** Proof of Proposition 2 is deferred to Appendix C.2. Any difference in prefix distributions directly translates into differences in hidden states $\mathbf{z}_n$ via Proposition 2. Thus, for the first layer of NoPE, positional distinguishability stems from $\boldsymbol{\beta}_n$ changing across $n$. This shows that positional encoding originates fundamentally from the sequence itself, merely extracted by the causal mask, rather than being injected solely through learned parameters. However, this perspective also reveals an intrinsic limitation: the model is permutation-invariant with respect to the prefix. As long as the current token is fixed, reordering earlier tokens leaves the contextual distribution (and thus $\mathbf{z}_n$) unchanged (Appendix D.1).

**Table 1:** Correlation metrics of probes under Polynomial Iteration of a 3-layer NoPE model. Abs. Pos. (L1): absolute position in layer 1, Rel. Pos. (L2, EoI): relative position (distance from EoI) in layer 2.

| Metric | Abs. Pos. (L1) | | Rel. Pos. (L2, EoI) | |
|--------|-------|------|-------|------|
| | Train | Test | Train | Test |
| $R^2$ | 0.975 | 0.864 | 0.891 | 0.814 |
| Pearson | 0.988 | 0.969 | 0.944 | 0.983 |
| Spearman | 0.992 | 0.975 | 0.944 | 0.995 |

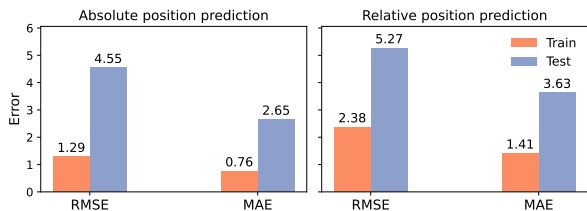

**Figure 5:** Probing errors under Polynomial Iteration of a 3-layer NoPE model. Left: absolute position prediction in layer 1. Right: relative position (distance from EoI) prediction in layer 2.

### 5.3 Probing Length Generalization in Iterative Tasks

Our earlier analysis suggested that NoPE encodes positions through simple statistical regularities but loses discriminability as $n$ grows. These results motivate us to test whether, in iterative tasks, NoPE indeed encodes positions in this way, and whether such encoding helps explain its limited LG.

To this end, we apply Linear Probing (Alain & Bengio, 2016; Haviv et al., 2022), i.e., training lightweight linear regressors on frozen hidden states $\mathbf{z}_n$ of a 3-layer Transformer. Probes predict absolute position from layer 1 and relative position (distance from EoI) from layer 2. As shown in Figure 5 and Table 1, probing errors (MSE) remain low and correlation coefficients high, confirming that positional signals are linearly decodable from $\mathbf{z}_n$, layer 1 captures absolute positions, while layer 2 encodes relative offsets, which algin with Kazemnejad et al. (2023). This validates the theoretical analysis that position information are monotonic and easy to extract in practice.

Table 1 further shows that correlation coefficients stay highly even in testing ($R^2 > 0.8$, Spearman/Pearson $> 0.9$), indicating that NoPE consistently preserves ordering and survives moderate extrapolation. However, Errors increase noticeably during testing (Figure 5), highlighting the effect of boundedness and decay, which misaligns with the task's computational structure since the task requires accurate positional signals to reliably locate $x_t$. This misalignment ultimately prevents NoPE from sustaining continuous length extrapolation.

> **Insight.** NoPE preserves positional consistency and achieves moderate extrapolation, but its bounded, decaying signals erase discriminability as $n$ grows, disturbing distance computation and breaking alignment with the task's computational structure, thus blocking further LG.

## 6 Aligning Inductive Biases for Length Generalization

Building on the previous analysis, we aim to reduce the two sources of misalignment: (i) structural bias from softmax attention and (ii) computational bias from PEs. To this end, we propose two augmentations: *(1) Logit controller.* Inspired by Chiang & Cholak (2022), we regulate attention logits to control the entropy of the attention distribution, reducing noise from irrelevant tokens and stabilizing anchor–target focus. *(2) Value-side relative PE.* We add a monotonic, bounded value-side distance PE with learnable scaling, ensuring consistent operation from training to extrapolation.

We suppose these augmentations align the model's inductive biases more closely with the computation required by iterative tasks. While heuristic, they illustrate how explicit alignment can help sustain the attention patterns and improve length generalization. We next detail the method, which we refer to as *ViPE*.

**Value-Side Relative Position Encoding** Let $\alpha_{ni} \geq 0$ be causal attention weights with $\sum_{i \leq n} \alpha_{ni} = 1$. For offsets $\delta_{ni} = n - i$, apply distance compression at test time:

$$\tilde{\delta}_{ni} = \begin{cases} \delta_{ni}, & \text{train}, \\ \delta_{ni}/s, & \text{fine-tuning/test}, \end{cases} \qquad s = \frac{L_{\max}^{\text{test}}}{L_{\max}^{\text{train}}}. \tag{7}$$

Define a value-side relative positional code

$$\mathbf{p}_{n-i} = W_p \tilde{\delta}_{ni} + b_p \in \mathbb{R}^{d_p}, \qquad \tilde{\mathbf{v}}_{ni} = \mathbf{v}_i \oplus \mathbf{p}_{n-i}, \qquad \mathbf{z}_n = \sum_{i \leq n} \alpha_{ni} \tilde{\mathbf{v}}_{ni}, \tag{8}$$

where $\mathbf{v}_i \in \mathbb{R}^{d_v}$ is the value vector at position $i$, $W_p \in \mathbb{R}^{d_p \times 1}$ and $b_p \in \mathbb{R}^{d_p}$ are learnable parameters, and $\oplus$ denotes concatenation, yielding $\tilde{\mathbf{v}}_{ni} \in \mathbb{R}^{d_v + d_p}$.

**Logit Rescaling for Attention Control**   Pre-softmax logits are adjusted as

$$\tilde{\ell}_{ni} = \lambda_{ni}\, \ell_{ni}^{\text{base}}, \qquad \lambda_{ni} = s \log(n)\, (1 + \mathbf{u}^\top \mathbf{k}_i). \tag{9}$$

where $\ell_{ni}^{\text{base}}$ is the original logit in Transformer (Vaswani et al., 2017), $\mathbf{k}_i$ is the key vector at position $i$, and $\mathbf{u}$ is a learned vector parameter. Here, $\log(n)$ suppresses length-driven entropy growth, $s$ restores contrast under distance compression, and $(1 + \mathbf{u}^\top \mathbf{k}_i)$ introduces a key-dependent rescaling factor. (See Appendix C.3 for heuristic derivation).

**Results.**   Figure 6 shows the accuracy of ViPE compared to other PEs, even with training restricted to short sequences, our method extrapolates to nearly twice the length with high accuracy, substantially outperforming other PEs. This supports that aligning inductive biases with task structure indeed sustains the attention pattern and improves length generalization.

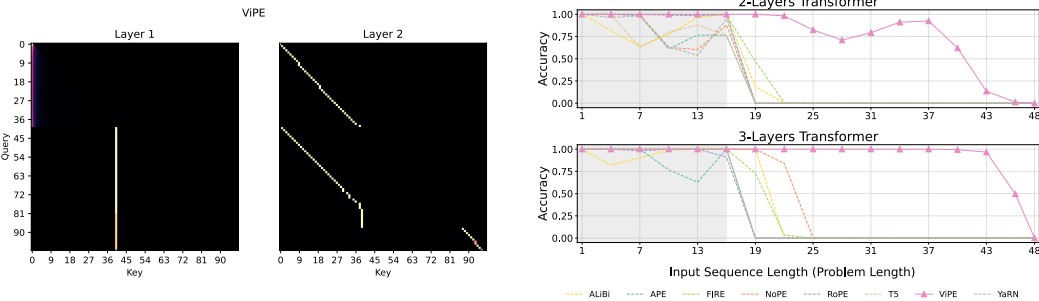

**Figure 6:** Left: Attention pattern under a 39 input-length (total sequence length 81) Polynomial Iteration task. Right: Train and test (input-length beyond 16) accuracy under Polynomial Iteration of all PEs (including our method ViPE).

**Table 2:** Accuracy of PEs and Mix RMonoAttn(MRMA) across CFQ and SCAN datasets.

| Dataset | APE | RoPE | NoPE | ViPE | MRMA | MRMA+ViPE |
|---|---|---|---|---|---|---|
| CFQ | 0.451 | 0.499 | 0.555 | **0.666** | 0.436 | 0.431 |
| SCAN | 0.000 | 0.021 | 0.132 | 0.150 | 0.162 | **0.293** |

**Further investigation on larger models and NLP-style reasoning tasks.**   Our main analysis focused on mechanistic interpretability using 2–3 layer, single-head Transformers under iterative tasks. While this setting enables fine-grained circuit-level understanding, it is natural to ask whether ViPE remains beneficial once we move beyond interpretable regimes—toward larger, multi-head architectures and natural-language-style compositional reasoning tasks, where the computation no longer admits clear mechanistic decomposition.

To examine this question, we follow prior work such as Chowdhury & Caragea (2023) and evaluate ViPE on SCAN and CFQ using a 4-head, 6-layer Transformer. Table 2 shows that ViPE achieves higher accuracy than commonly used PEs on both datasets. Moreover, when incorporated into the Mix RMonoAttn architecture of Chowdhury & Caragea (2023), ViPE further improves SCAN accuracy from 0.162 to 0.293. While these results do not aim to provide a full mechanistic account—larger architectures are harder to analyze—they suggest that the alignment principles studied in our small-model experiments may continue to be useful in more realistic settings.

Overall, this exploratory extension indicates that ViPE, despite being motivated by insights from iterative tasks, may have broader applicability. These observations motivate future work on connecting mechanistic alignment with large-scale architectures and more complex reasoning domains.

## 7 CONCLUSION

We examined length generalization (LG) of Transformers through the lens of alignment between task structure and model inductive bias. Across iterative tasks, we systematically analyzed different positional encodings (PEs) and found that although Transformers can partially align, misalignment persists: structural bias from softmax attention and computational biases from PEs destabilize LG under extrapolation, causing accuracy to collapse. Notably, while NoPE shows the strongest potential for LG—supported by implicit positional signals in hidden-state statistics and contextual token distributions—these signals fade with length, leaving its encoding misaligned with task requirements. Guided by this mechanistic analysis, we introduced a lightweight enhancement, value-side relative coding with logit rescaling, which sustains iterative computation and improves LG. Our findings suggest that aligning inductive bias with computational structure is key to robust LG, and provide concrete directions for future PE design.

**Limitation**. While we have identified the key factors that determine length generalization in iterative tasks, we have not yet investigated whether these factors can be applied or extended to other tasks with similar structures (e.g., other types of mathematical reasoning problems). Second, regarding the collapse of computation patterns under different PEs, each PE may induce its own unique computational pathway. Although these can be abstracted into the broader notion of computational inductive bias, the precise nature of the erroneous information transmitted by each PE—and how this leads to deviations from the computation embodied by the attention pattern—still requires deeper theoretical analysis. Finally, this paper merely provides an initial attempt to align inductive biases, showing that better task–model alignment can indeed improve generalization. Developing a positional encoding that generalizes robustly across a wide range of tasks will require further investigation.

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

# A  EXTENDED PRELIMINARIES

## A.1  ORIGINAL ITERATIVE TASKS

**Definition.**  Cabannes et al. (2024) define the iterative task problems that are naturally solved by an iterative algorithm over an input sequence $X = (x_1, \dots, x_L)$ with an internal state $s_t$ updated as

$$s_0 = \text{Init}, \qquad s_t = f(s_{t-1}, x_t), \quad t = 1, \dots, L, \tag{10}$$

for some update function $f$. An iterative task asks the model to produce either the final state $s_L$ or the whole trajectory $(s_1, \dots, s_L)$ induced by equation 10.

**Why transformers struggle without CoT.**  Under next-token prediction without chain-of-thought (CoT), a depth-$D$ transformer must compress the entire multi-step computation into a bounded number of cross-operations before emitting the answer token (Li et al., 2024; Feng et al., 2023). For many iterative tasks, the mapping $X \mapsto s_L$ behaves like a deep composition (e.g., repeated affine or multiplicative updates), which induces long-range, high-order interactions in $X$. With bounded depth, this is provably or empirically hard to learn and generalize. In contrast, with CoT the model emits the intermediate states $(s_1, \dots, s_L)$ as tokens, externalizing the recursion: at each step it only needs to read $(x_t, s_{t-1})$ and apply a local update to produce $s_t$. This aligns with the autoregressive inductive bias and effectively endows the model with the ability to perform serial computation across steps.

**Canonical examples.**  Following Cabannes et al. (2024), we recall three representative instances of iterative tasks.

1. **Polynomial iteration.** Inputs are elements $x_t \in \mathbb{F}_p = \mathbb{Z}/p\mathbb{Z}$ for a prime $p$. The state is initialized at $s_1 = x_1$ and updated by a fixed bivariate function $f : \mathbb{F}_p \times \mathbb{F}_p \to \mathbb{F}_p$:

$$s_t = f(s_{t-1}, x_t) \bmod p.$$

   This family encompasses a wide range of iterative dynamics, since $f$ can mix additive and multiplicative interactions. A common affine instance is

$$s_t = (s_{t-1} \cdot x_t + \beta) \bmod p,$$

   which already requires the model to compose multiplicative and additive operations.

2. **Parity.** The parity problem arises as a degenerate instance of polynomial iteration when $p = 2$ and $f(s, x) = s + x$. In that case,

$$s_t = (s_{t-1} + x_t) \bmod 2,$$

   so $s_t$ simply records the parity (even/odd) of the prefix sum $\sum_{i=1}^{t} x_i$. Despite its apparent simplicity, parity highlights the difficulty of long-range interactions for depth-limited transformers.

3. **Binary.** The copying problem is the simplest instance of an iterative scheme. Each token is $x_t \in \{0, 1\}$, with initial state $s_0 = 0$, and the update rule

$$s_t = f(s_{t-1}, x_t) = x_t.$$

   That is, the state at each step simply reproduces the current input.

## A.2  ORIGINAL ITERATION HEADS

Following Cabannes et al. (2024), an iteration head refers to a specific two-layer attention pattern that allows a transformer to implement iterative algorithms by chain-of-thought reasoning. We briefly recall the key elements here for completeness.

1. **First attention head:** locates the end-of-input (EoI) token via a query-key mechanism ("Are you EoI?"), thus retrieving its positional code.

2. **Second attention head:** uses this positional cue to retrieve the current input token $x_t$, while residual connections carry forward the previous state $s_{t-1}$.

3. **MLP:** computes the update $s_t = f(s_{t-1}, x_t)$, leveraging universal approximation.

This theoretical circuit is agnostic to the choice of $f$ and thus applicable to any iterative task (Parity, Polynomial, Copy). In words, the first attention head locates the EoI position, the second retrieves the current input $x_t$ and previous state $s_{t-1}$, and the MLP implements the update $s_t = f(s_{t-1}, x_t)$.

## A.3 LENGTH NOTIONS AND EXTRAPOLATION

**Notation.** For a sequence $x$, let $L_{\text{in}}(x)$ denote the *input length* (problem length), and let $L_{\text{tot}}(x)$ denote the *total sequence length* fed to the model, including reasoning steps and special tokens. Positions are indexed by $p \in \{1, \ldots, L_{\text{tot}}(x)\}$. Let

$$L_{\text{in}}^{\text{max,train}}, \quad L_{\text{tot}}^{\text{max,train}}, \quad p^{\text{max,train}}$$

be the maximum input length, total length, and absolute position index encountered during training.

**Two notions of length extrapolation.** We distinguish two common usages:

- **Input-length extrapolation** (input-o.o.d.): a test instance $x$ satisfies $L_{\text{in}}(x) > L_{\text{in}}^{\text{max,train}}$.
- **Total-length (position) extrapolation** (total-o.o.d.): a test instance (or a prediction within it) satisfies $L_{\text{tot}}(x) > L_{\text{tot}}^{\text{max,train}}$ or the evaluated token lies at a position $p > p^{\text{max,train}}$ (i.e., the model is queried at an unseen absolute index).

**Coupling in iterative tasks.** In iterative reasoning tasks (e.g., Polynomial Iteration, Parity, Binary Copy), the sequence assembled for the model typically has an affine dependence on the problem size; in our setup,

$$L_{\text{tot}}(x) \approx 2\, L_{\text{in}}(x) + 3,$$

so input and total length are tightly coupled though not identical.

## A.4 RPE-INDUCED DISTANCE-ATTENTION BIAS

We briefly summarize the relative positional encodings used in this paper and fix notation. Let $\delta = n - i$ denote the relative distance between a query position $n$ and a key position $i$, and let $\ell_{ni}$ be the attention logit.

**RoPE (Su et al., 2024).** Pair coordinates into $(2m-1, 2m)$ with per-band angle $\theta_m$ and define the rotation $R(n) = \text{diag}(R_{\theta_1 n}, \ldots, R_{\theta_M n})$, where $R_\theta = \left(\begin{smallmatrix} \cos\theta & -\sin\theta \\ \sin\theta & \cos\theta \end{smallmatrix}\right)$. Applying RoPE to queries/keys gives

$$\ell_{ni}^{\text{RoPE}} = \langle R(n)\mathbf{q},\ R(i)\mathbf{k} \rangle = \langle \mathbf{q},\ R(\delta)\mathbf{k} \rangle = \sum_{m=1}^{M} \langle \mathbf{q}^{(m)}, R_{\theta_m \delta}\mathbf{k}^{(m)} \rangle.$$

Thus the logit depends on the relative distance $\delta$ via band-wise rotations. When multiple frequencies are mixed, heterogeneous phases across $m$ can reduce the aggregate similarity for large $|\delta|$ (effective distance–attention attenuation in dot-product)] (Su, 2021).

**ALiBi (Press et al., 2021).** ALiBi adds a head-specific linear penalty in distance:

$$\ell_{ni}^{\text{ALiBi}} = \langle \mathbf{q}_n, \mathbf{k}_i \rangle - \alpha_h\, |\delta|, \qquad \alpha_h > 0,$$

explicitly favoring closer tokens.

**FIRE (Li et al., 2023).** FIRE augments logits with a kernel of relative distance,

$$\ell_{ni}^{\text{FIRE}} = \langle \mathbf{q}_n, \mathbf{k}_i \rangle + g(\delta),$$

where $g(\cdot)$ is chosen/learned; Though attention does not necessarily decay with distance, the introduced bias is still only related to $|\delta|$.

**Implication for our setting.** In iterative tasks, stable attention to anchors (e.g., BoS/EoI) and targets $x_t$ is desirable to repeat the same local update across longer inputs. Distance-coupled logits (ALiBi/FIRE and the phase-dispersion effect in RoPE) can attenuate such anchor/target attention as $|\delta|$ grows, which is the "distance–attention bias" we refer to in the main text.

# B EXPERIMENTAL DETAILS

**Datasets and tasks** We evaluate three synthetic reasoning tasks: Binary Copy, Parity, and Polynomial Iteration (Cabannes et al., 2024; Zhou et al., 2023). Each dataset is procedurally generated to ensure full coverage of input lengths. For training, input lengths $L_{\text{in}}$ are uniformly sampled from 1 to 16, yielding full sequence lengths between 5 and 35. For testing, input lengths are extended to $17 \leq L_{\text{in}} \leq 48$, corresponding to full sequence lengths from 37 to 99. Each task contains $N_{\text{train}} = 32{,}768$ and $N_{\text{test}} = 65{,}536$ samples, with 2,048 samples per input length.

**Table 3:** Examples of iterative tasks.

| Task | Explanation | Example (Input / Output) |
|------|-------------|--------------------------|
| Binary Copy | Copy a repeated binary sequence in order. | Input: `[BoS, 1, 0, 1, 0, 0, EoI]`
Output: `[1, 0, 1, 0, 0, EoS]` |
| Polynomial | Running update $e_i = (e_{i-1} \cdot x_i) + 1 \bmod 5$, output each $e_i$. | Input: `[BoS, 1, 2, 3, 4, EoI]`
Output: `[1, 3, 0, 1, EoS]` |
| Parity | Special Polynomial iteration with $p = 2$ and update rule $s_t = (s_{t-1} + x_t) \bmod 2$. | Input: `[BoS, 1, 1, 1, 0, 1, EoI]`
Output: `[1, 0, 1, 1, 0, EoS]` |

**Models and encodings** We implement small Transformers with PreNorm attention and MLP blocks. We mainly consider 1-head, 2-layer and 1-head, 3-layer models. We compare eight positional encodings: Sinusoidal APE, NoPE, RoPE, ALiBi, T5 Relative Bias, FIRE, and YaRN. For FIRE, we additionally search hidden sizes $\{32, 64\}$; for T5, we search bucket numbers $\{24, 32\}$.

**Training configuration and hyperparameters** Models are trained with Adam optimizer ($lr = 3 \times 10^{-4}$), batch size 256, for 2000 epochs. We evaluate embedding dimensions $\{32, 128\}$ and both 2-layer and 3-layer settings. Each experiment is repeated with 8 random seeds $\{0, 42, 123, 2025, 7811, 9527, 13579, 23343\}$. All experiments are run on NVIDIA H100 GPUs.

## B.1 ADDITIONAL EXPERIMENTTAL SETTINGS

Unless otherwise stated, experiments follow the general setup described in Appendix B. We highlight here the special configurations for each experiment.

**Exp. 1: Attention maps.** We use embedding sizes $\{32, 128\}$ and sample uniformly from input lengths 1–48 across both training and testing to visualize the emergence of attention patterns. In addition to the PEs analyzed in the main text, we also examine *Base* (learned APE) and *VRoPE*, the latter being a variant inspired by ViPE where the linear transformation is replaced with a rotation on the value ($V$) stream. Neither of these are included in the main text: Base trivially lacks generalization ability, and VRoPE is exploratory and overlaps with ViPE. We report only their attention maps for reference.

**Exp. 2: Overall accuracy.** We fix embedding size 128 and evaluate the best test accuracy across all random seeds, while ensuring that the corresponding training accuracy exceeds 0.85.

**Exp. 3: Common PE accuracy.** We reuse the 3-layer models trained in Exp. 2. Evaluation is performed on 20,480 test samples, with the positions of reasoning tokens constrained to remain within the maximum training sequence length 35.

**Exp. 4: Linear probing.** We use the 3-layer NoPE model trained in Exp. 2. Probes are trained on all training data, and tested on 12,000 samples drawn from the test set.

**Exp. 5: Statistics of hidden states.** We use the 3-layer NoPE model trained in Exp. 2. For analysis, we compute the mean and variance of the hidden states in the first layer and plot their trajectories across positions.

**Exp. 6: Average accuracy on additional synthetic tasks.** We follow the same experimental configuration as in Exp. 2 but additionally evaluate on four synthetic reasoning tasks: *Count*, *Reverse*, *Mode*, and *Sort*.

**Exp. 7: Average accuracy on iterative tasks with larger models.** To investigate whether larger and deeper Transformers improve performance on iterative tasks, we experiment with models using 2 or 4 attention heads and 4–6 layers. The remaining hyperparameters follow those in Exp. 2 unless otherwise specified.

**Exp. 8: Total accuracy on SCAN and CFQ with larger models.** To explore the generalization behavior of different PEs on natural-language-style reasoning tasks, we evaluate a larger model configured with 4 attention heads, 6 layers, and an embedding size of 256. Using this fixed architecture, we measure total accuracy on the SCAN and CFQ benchmarks to examine how various PEs behave when both model capacity and task difficulty increase.

## C PROOFS

### C.1 STATISTICAL ENCODING UNDER NOPE

We provide the proof of Proposition 1.

**Notation and setup.** We analyze the first attention layer under the simplifying assumption of uniform weights $\alpha_{ni} = \frac{1}{n}$ for $i \leq n$, i.e. the output at position $n$ is the arithmetic mean of the first $n$ value vectors. Let $i$ index token positions, $j$ index embedding dimensions, and $d$ denote the embedding size.

For each ordinary token's value $\mathbf{v_i}$ and coordinate $j = 1, \ldots, d$, we assume i.i.d. Gaussian entries

$$v_{i,j} \overset{\text{i.i.d.}}{\sim} \mathcal{N}(\mu, \sigma^2), \qquad \mathbf{v}_i := (v_{i,1}, \ldots, v_{i,d}) \in \mathbb{R}^d. \tag{11}$$

For the special BoS token, we allow a potentially different distribution:

$$b_j \overset{\text{i.i.d.}}{\sim} \mathcal{N}(\mu_2, \sigma_2^2), \qquad \mathbf{v}_{\text{BoS}} := (b_1, \ldots, b_d) \in \mathbb{R}^d. \tag{12}$$

The causal average at position $n$ is then

$$\mathbf{z}_n = \frac{\mathbf{v}_{\text{BoS}} + \sum_{i=1}^{n-1} \mathbf{v}_i}{n}, \qquad z_{n,j} \text{ denoting its } j\text{-th coordinate.} \tag{13}$$

The prefix consists of the BoS token plus the first $n-1$ ordinary tokens. Here $x_i$ denotes the $i$-th input token, $\mathbf{v}_i$ its associated value vector, and $\mathbf{z}_n$ the attention output at position $n$.

Since coordinates are i.i.d. and $d$ is large, empirical averages across dimensions concentrate around expectations:

$$\frac{1}{d} \sum_{j=1}^{d} z_{n,j} \approx \mathbb{E}[z_{n,j}], \qquad \frac{1}{d} \sum_{j=1}^{d} \left(z_{n,j} - \frac{1}{d} \sum_{k=1}^{d} z_{n,k}\right)^2 \approx \text{Var}(z_{n,j}). \tag{14}$$

**Claim 1 (Hidden-state mean).** The across-dimensions hidden-state mean converges to a value depending on $n$:

$$\frac{1}{d} \sum_{j=1}^{d} z_{n,j} \xrightarrow[d \to \infty]{\mathbb{P}} \mu + \frac{\mu_2 - \mu}{n}. \tag{15}$$

In practice, for large $d$ this concentration is expressed as

$$\frac{1}{d} \sum_{j=1}^{d} z_{n,j} \approx \mu + \frac{\mu_2 - \mu}{n}. \tag{16}$$

*Proof.* (1) By linearity of expectation,

$$z_{n,j} = \frac{1}{n}\left(b_j + \sum_{i=1}^{n-1} v_{i,j}\right), \tag{17}$$

$$\mathbb{E}[z_{n,j}] = \frac{1}{n}\left(\mathbb{E}[b_j] + \sum_{i=1}^{n-1} \mathbb{E}[v_{i,j}]\right) = \frac{1}{n}\left(\mu_2 + (n-1)\mu\right) = \mu + \frac{\mu_2 - \mu}{n}. \tag{18}$$

(2) By the weak law of large numbers applied across coordinates $j$,

$$\frac{1}{d}\sum_{j=1}^{d} z_{n,j} \xrightarrow[d\to\infty]{\mathbb{P}} \mathbb{E}[z_{n,j}] = \mu + \frac{\mu_2 - \mu}{n}, \tag{19}$$

This completes the proof of Claim 1. $\square$

**Claim 2 (Hidden-state variance).** The across-dimensions hidden-state variance concentrates at a value depending on $n$:

$$\frac{1}{d}\sum_{j=1}^{d}\left(z_{n,j} - \frac{1}{d}\sum_{k=1}^{d} z_{n,k}\right)^2 \xrightarrow[d\to\infty]{\mathbb{P}} \frac{\sigma_2^2 + (n-1)\sigma^2}{n^2}. \tag{20}$$

In practice, for large $d$ this concentration is expressed as

$$\frac{1}{d}\sum_{j=1}^{d}\left(z_{n,j} - \frac{1}{d}\sum_{k=1}^{d} z_{n,k}\right)^2 \approx \frac{\sigma_2^2 + (n-1)\sigma^2}{n^2}. \tag{21}$$

*Proof.* By independence across $b_j$ and $\{v_{i,j}\}_i$,

$$\mathrm{Var}(z_{n,j}) = \mathrm{Var}\left(\frac{1}{n}\left(b_j + \sum_{i=1}^{n-1} v_{i,j}\right)\right) \tag{22}$$

$$= \frac{1}{n^2}\left(\mathrm{Var}(b_j) + \sum_{i=1}^{n-1}\mathrm{Var}(v_{i,j})\right) \tag{23}$$

$$= \frac{1}{n^2}\left(\sigma_2^2 + (n-1)\sigma^2\right). \tag{24}$$

Then by the weak law of large numbers across coordinates $j$,

$$\frac{1}{d}\sum_{j=1}^{d}(z_{n,j} - \frac{1}{d}\sum_{k} z_{n,k})^2 \xrightarrow[d\to\infty]{\mathbb{P}} \mathrm{Var}(z_{n,j}) = \frac{\sigma_2^2 + (n-1)\sigma^2}{n^2},$$

This completes the proof of Claim 2. $\square$

**Claim 3 (Mean of adjacent hidden-state difference).** The across-dimensions mean difference between adjacent positions concentrates at

$$\frac{1}{d}\sum_{j=1}^{d}(z_{n+1,j} - z_{n,j}) \xrightarrow[d\to\infty]{\mathbb{P}} -\frac{\mu_2 - \mu}{n(n+1)}. \tag{25}$$

In practice, for large $d$ this is expressed as

$$\frac{1}{d}\sum_{j=1}^{d}(z_{n+1,j} - z_{n,j}) \approx -\frac{\mu_2 - \mu}{n(n+1)}. \tag{26}$$

*Proof.* By direct algebra,

$$z_{n+1,j} - z_{n,j} = \frac{b_j + \sum_{i=1}^{n} v_{i,j}}{n+1} - \frac{b_j + \sum_{i=1}^{n-1} v_{i,j}}{n} \tag{27}$$

$$= \frac{v_{n,j}}{n+1} - \frac{b_j + \sum_{i=1}^{n-1} v_{i,j}}{n(n+1)} = \frac{v_{n,j} - z_{n,j}}{n+1}. \tag{28}$$

Taking expectations and using the expression for $\mathbb{E}[z_{n,j}]$,

$$\mathbb{E}[z_{n+1,j} - z_{n,j}] = \frac{\mu - \left(\mu + \frac{\mu_2 - \mu}{n}\right)}{n+1} = -\frac{\mu_2 - \mu}{n(n+1)}. \tag{29}$$

Finally, by the weak law of large numbers across dimensions,

$$\frac{1}{d} \sum_{j=1}^{d} (z_{n+1,j} - z_{n,j}) \xrightarrow[d\to\infty]{\mathbb{P}} \mathbb{E}[z_{n+1,j} - z_{n,j}], \tag{30}$$

This completes the proof of Claim 3. $\qquad\square$

Thus, combining the above results, Proposition 1 is proved.

## C.2 CONTEXTUAL TOKEN DISTRIBUTIONS UNDER ORIGINAL SEQUENCES

**Notation and setup.** Following the above setting, We analyze a single head with causal masking. The value path is treated as linear, so any fixed per-layer linear map (embedding and $W_V$) is absorbed into the category basis $V$.

We model the sequence as drawn from a fixed set of $C$ content categories. Let $\mathcal{C} = \{1, \ldots, C\}$ and $S = (c_1, \ldots, c_L)$ with $c_n \in \mathcal{C}$. Each category $c$ has a value embedding $\mathbf{v}_c \in \mathbb{R}^d$, collect them as $V = [\mathbf{v}_1, \ldots, \mathbf{v}_C] \in \mathbb{R}^{d \times C}$. Each position $i$ in the sequence carries a value vector $\mathbf{v}_i$, which is chosen from the embedding matrix $V$ according to its category $c_i$, i.e. $\mathbf{v}_i = \mathbf{v}_{c_i}$.

Let $\alpha_{ni} \geq 0$ denote causal attention weights ($i \leq n$) with row-stochastic normalization $\sum_{i \leq n} \alpha_{ni} = 1$. Define the index set $I_c(n) := \{ i \leq n : c_i = c \}$. We interpret

$$\beta_{c,n} := \sum_{i \in I_c(n)} \alpha_{ni} \tag{31}$$

as the normalized attention mass allocated to category $c$ within the prefix $1{:}n$. Thus $\boldsymbol{\beta}_n = (\beta_{1,n}, \ldots, \beta_{C,n}) \in \Delta^{C-1}$ represents the contextual token distribution over categories up to position $n$.

**Claim.** The one layer attention output is a linear embedding of the contextual token distribution:

$$\mathbf{z}_n = V\boldsymbol{\beta}_n. \tag{32}$$

*Proof.* By definition,

$$\mathbf{z}_n = \sum_{i \leq n} \alpha_{ni}\, \mathbf{v}_{c_i} = \sum_{c=1}^{C} \left( \sum_{i \in I_c(n)} \alpha_{ni} \right) \mathbf{v}_c = \sum_{c=1}^{C} \beta_{c,n}\, \mathbf{v}_c = V\boldsymbol{\beta}_n. \tag{33}$$

$\qquad\square$

Based on the claim above, we conclude the proof of Proposition 2.

**Corollaries.**

- Rank condition: if $C \leq d$ and $V$ full rank, then $\mathbf{z}_n$ uniquely determines $\boldsymbol{\beta}_n$.
- General case: if $C > d$, $\mathbf{z}_n$ compresses $\boldsymbol{\beta}_n$ but still linearly reflects distributional differences.
- Positional signal: for $m \neq n$, if $\boldsymbol{\beta}_n \neq \boldsymbol{\beta}_m$, then
$$\mathbf{z}_n - \mathbf{z}_m = V(\boldsymbol{\beta}_n - \boldsymbol{\beta}_m), \tag{34}$$
hence any contextual distribution difference is preserved in hidden space.

## C.3 SCALED COEFFICIENT IN ATTENTION LOGITS

### C.3.1 SECOND–ORDER TAYLOR EXPANSION OF $\delta(s)$

The following derivation is heuristic: it relies on simplified geometric assumptions and is intended to provide intuition for how positional scaling affects attention logits, rather than a rigorous proof.

Since increasing $s$ makes positional encodings more similar at longer lengths, the logits $\langle \mathbf{q}, \mathbf{k} \rangle$ between different positions also become closer, causing attention drift. To counter this, one must preserve the relative differences between attention logits, namely the dot-products of $\mathbf{q}$ and $\mathbf{k}$. In many formulations, position is encoded in transformers through a rotation angle rather than through the norm of the vectors. We assume (i) $\|\mathbf{q}\| = \|\mathbf{k}\|$, and (ii) $\theta$ denotes the baseline angle between $\mathbf{q}$ and its target key, while a unit positional step ($\delta = 1$) contributes an additional very small rotation $w$.

**Original form** With equal norms, the logit difference arises from the cosine term:

$$\delta = \cos\theta - \cos(\theta + w). \tag{35}$$

**Taylor expansion with remainder.** Expanding around $w = 0$ up to second order, with Lagrange remainder:

$$\cos(\theta+w) = \cos\theta - w\sin\theta - \frac{w^2}{2}\cos\theta + R_3(w), \qquad R_3(w) = -\frac{w^3}{6}\sin(\theta+\xi w), \ \xi \in (0,1). \tag{36}$$

Subtracting from $\cos\theta$ gives

$$\delta = w\sin\theta + \frac{w^2}{2}\cos\theta + R_3(w), \qquad |R_3(w)| \le \frac{|w|^3}{6}. \tag{37}$$

**Approximation for small $w$.** When $|w| \ll 1$ (empirically true under training lengths), the cubic remainder is negligible, yielding

$$\delta \approx w\sin\theta + \frac{w^2}{2}\cos\theta. \tag{38}$$

**Inference regime ($s > 1$).** When scaling is applied, a unit positional step is compressed to $w/s$. Substituting $w \mapsto w/s$ in equation 38 yields

$$\delta(s) \approx \frac{w}{s}\sin\theta + \frac{w^2}{2s^2}\cos\theta. \tag{39}$$

Thus the leading term decays as $1/s$, the quadratic correction as $1/s^2$, and the overall logit difference shrinks with increasing $s$, explaining the drift observed at longer lengths.

### C.3.2 RATIONAL SCALING CANCELS $1/s$ AND $1/s^2$ WHILE PRESERVING $\delta(s)$

From the second–order expansion we have

$$\delta \approx A + B, \qquad \delta(s) \approx \frac{A}{s} + \frac{B}{s^2}, \tag{40}$$

where

$$A := w\sin\theta, \qquad B := \frac{w^2}{2}\cos\theta. \tag{41}$$

We seek a rational scaling factor

$$g(s) = a\,s + \frac{b}{s} + c \tag{42}$$

such that $g(s)\,\delta(s) \approx \delta$ while canceling the $1/s$ and $1/s^2$ terms. Expanding:

$$g(s)\,\delta(s) = \left(as + \frac{b}{s} + c\right)\left(\frac{A}{s} + \frac{B}{s^2}\right) = aA + \frac{aB + cA}{s} + \frac{bA + cB}{s^2} + \frac{bB}{s^3}. \tag{43}$$

The constraints are

$$aA = A + B, \qquad \text{(match constant)} \tag{44}$$

$$aB + cA = 0, \qquad \text{(cancel } 1/s) \tag{45}$$

$$bA + cB = 0, \qquad \text{(cancel } 1/s^2). \tag{46}$$

Solving equation 44–equation 46 yields

$$a = 1 + \frac{B}{A}, \qquad c = -\frac{aB}{A}, \qquad b = \frac{aB^2}{A^2}. \tag{47}$$

For compactness, let

$$r := \frac{B}{A} = \frac{w}{2} \cot \theta. \tag{48}$$

Then

$$a = 1 + r, \qquad b = a\,r^2, \qquad c = -a\,r, \tag{49}$$

so the final scale is

$$g(s) \;=\; (1+r)\Big(s - r + \frac{r^2}{s}\Big), \qquad r = \frac{w}{2} \cot \theta. \tag{50}$$

With this choice,

$$g(s)\,\delta(s) \;=\; \delta \;+\; O\big(s^{-3}\big), \tag{51}$$

i.e. the constant matches the training regime ($s = 1$), and the $1/s$ and $1/s^2$ dependencies are canceled. Only the $1/s^3$ residual remains.

**Practical considerations.** The derivation assumes $\|\mathbf{q}\| = \|\mathbf{k}\|$ and ignores cross-key variation. In practice, keys differ and $\|\mathbf{q}\| \neq \|\mathbf{k}\|$, so we add a correction factor $(1 + \mathbf{u}^\top \mathbf{k})$ to $g(s)$, allowing adaptation to local key statistics. Moreover, we found empirically that simpler forms such as $g(s) = s$ or $g(s) = as + b$ yield more stable training, while explicit $1/s$ or $s^2$ terms can over–concentrate attention and cause gradient explosion. Thus the form equation 50 provides useful intuition, but implementation could be adjusted.

Finally, even if distance scaling is compensated by $g(s)$, irrelevant tokens still contribute to the softmax denominator, causing entropy growth with sequence length $n$. Following prior analyses of entropy growth in softmax attention (Chiang & Cholak, 2022), a logarithmic factor $\log n$ can be applied for entropy control. In our experiments, we combine rational scaling with $\log n$ to stabilize attention across lengths, thus the final logit rescaling factor is $s \log(n)\,(1 + \mathbf{u}^\top \mathbf{k}_i)$.

# D  RESULTS DETAILS

## D.1  CONTEXTUAL TOKEN DISTRIBUTIONS IN CASES.

Based on Proposition 2, one can easily judge which input sequences are distinguishable under NoPE, for one layer attention with uniform weights $\alpha_{ni} = \frac{1}{n}, i \leq n$:

- $[1, 1, 1, 1]$: The context distribution is identical at every position (all tokens are "1"), so no positional differences can be encoded.
- $[\text{BoS}, 1, 1, 1, 1]$: The proportion of BoS decreases while the proportion of "1" increases with $n$, so absolute positions are distinguishable.
- $[1, 2, 3, 4, 1, 2, 3, 4]$: The two occurrences of token "4" have identical prefix distributions, hence they cannot be distinguished.
- $[\text{BoS}, 1, 2, 3, 4, 1, 2, 3, 4]$: With BoS included, the prefix distributions for the two "4" tokens differ, making them distinguishable.

These illustrative cases show that the discriminative power of a single-layer mean-attention model is fully determined by whether the prefix distributions $\boldsymbol{\beta}_n$ change across positions.

However, this perspective also reveals a more general symmetry of single-layer NoPE: it is permutation invariant with respect to the ordering of the prefix. For instance, compare $[\text{BoS}, 1, 2, 3, 4]$ and $[\text{BoS}, 2, 1, 3, 4]$. For the current token "4" the prefix distributions are identical in both cases, so their representations $\mathbf{z}_n$ also coincide. In other words, as long as the current token is fixed, a single-layer NoPE is insensitive to permutations of its prefix, the output remains unchanged.

## D.2 EXPERIMENT RESULTS

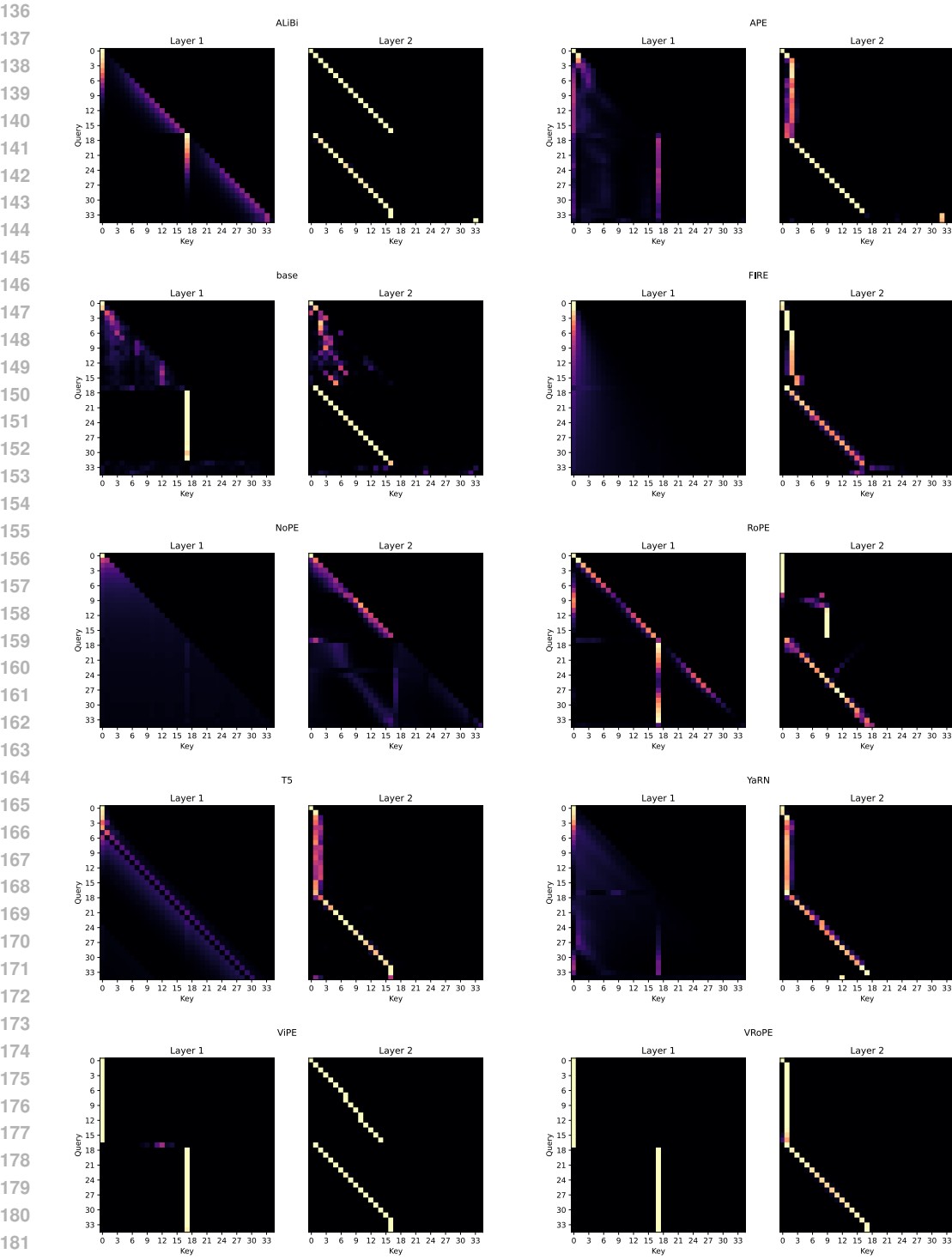

**Figure 7:** Anchor-based attention patterns for 2-layer models with different PEs on iterative tasks. Besides the PEs in the main text, we also check Learned Positional Embeddings (Base) (Brown et al., 2020) and VRoPE (rotary encoding on values), but omit them since Base trivially fails to generalize and VRoPE is only a simple ViPE-inspired variant with limited insights.

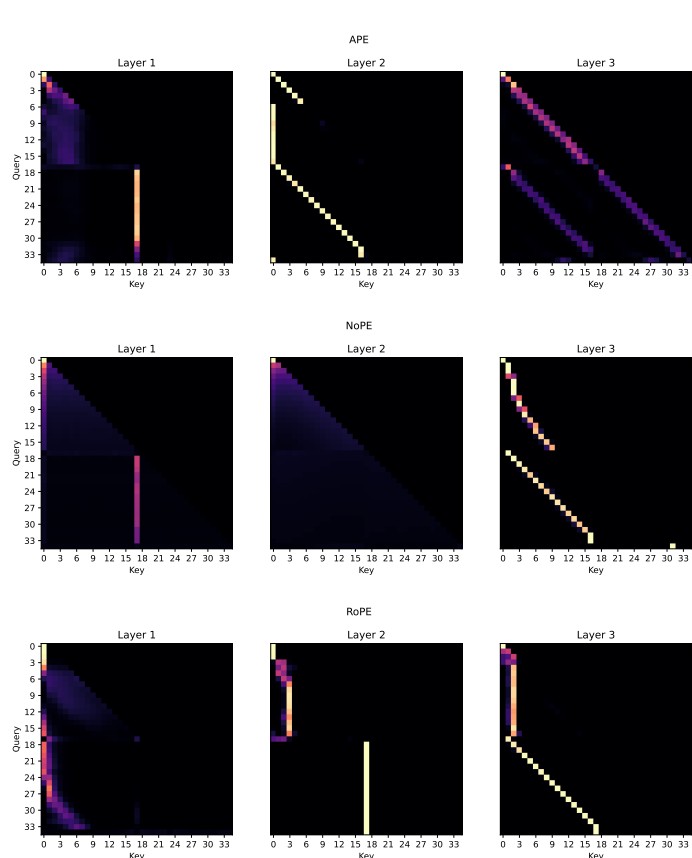

**Figure 8:** Anchor-based attention maps for 3-layer models with APE, NoPE, RoPE

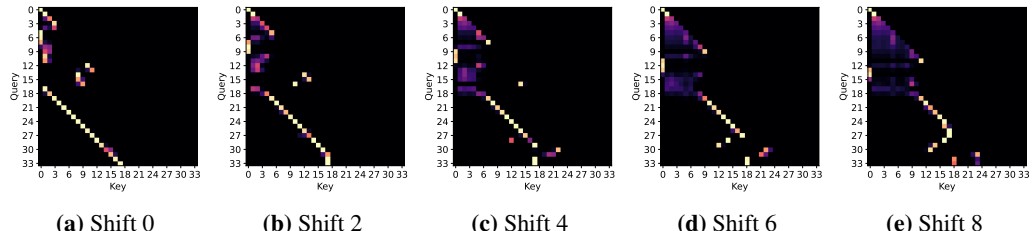

**(a)** Shift 0     **(b)** Shift 2     **(c)** Shift 4     **(d)** Shift 6     **(e)** Shift 8

**Figure 9:** Attention maps of RoPE under different BoS shifts (i.e., shifting all tokens uniformly so that their relative distance to BoS is offset by -2, -4, -6, etc.). The attention for $x_t$ shifts rightward as the offset increases.

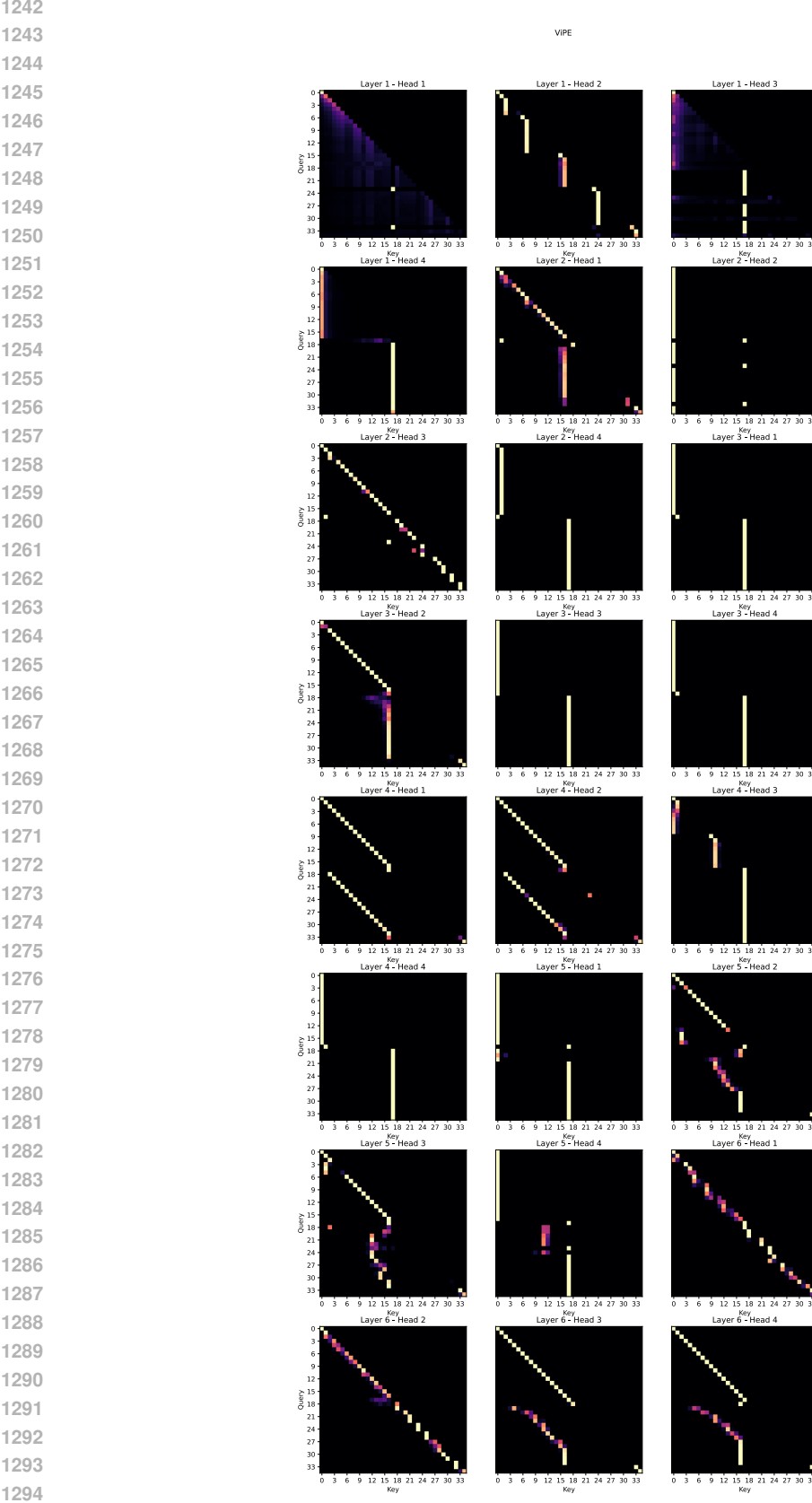

**Figure 10:** Attention maps of ViPE for 4-heads, 6-layers models on Polynomial Iteration.

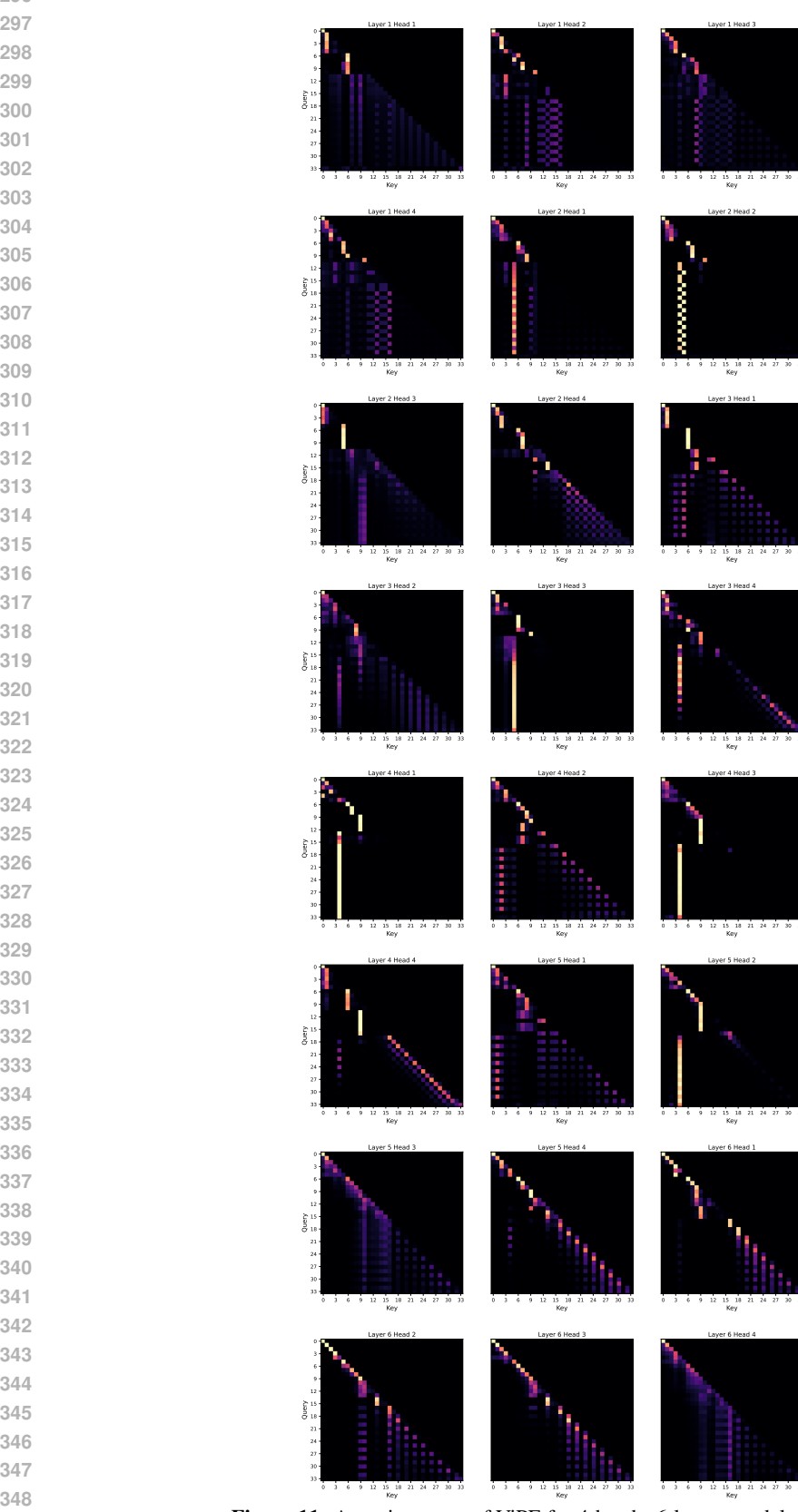

**Figure 11:** Attention maps of ViPE for 4-heads, 6-layers models on Scan.

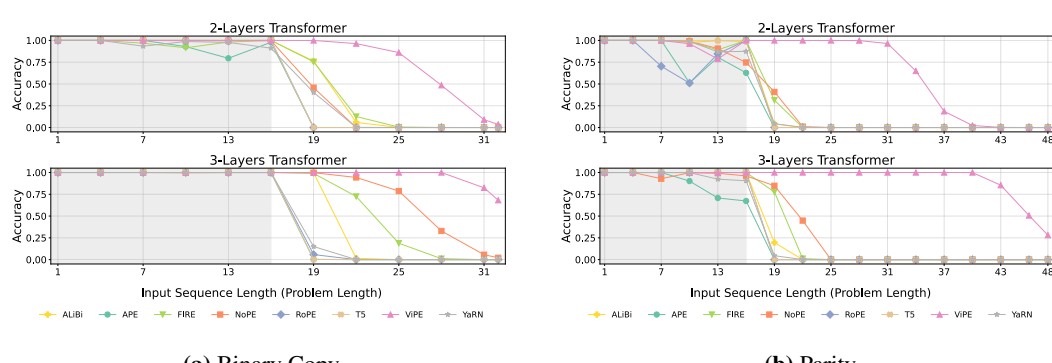

(a) Binary Copy

(b) Parity

**Figure 12:** Best accuracy for Binary Copy and Parity on 2-layers and 3-layers models with 1 head.

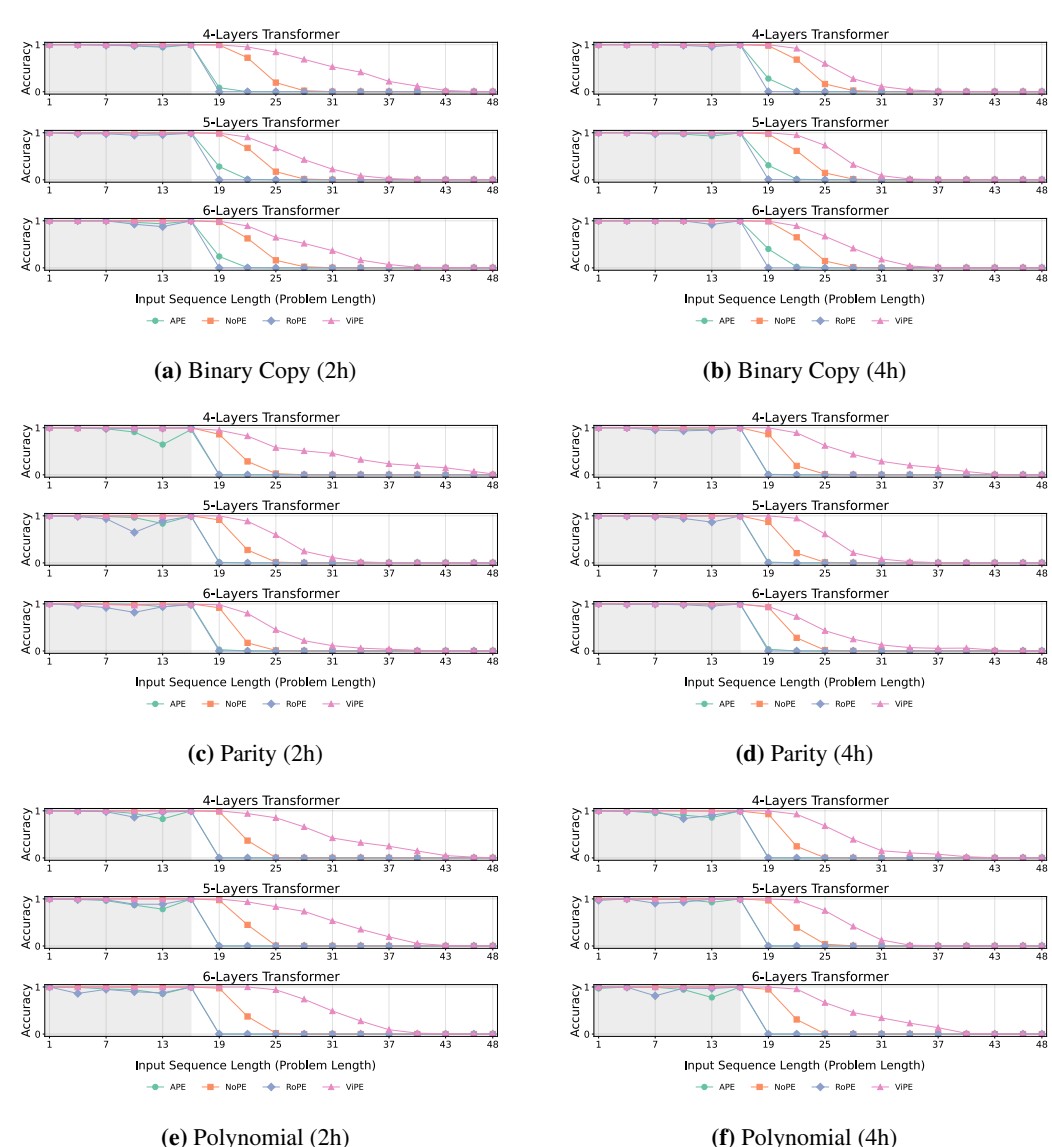

(a) Binary Copy (2h)

(b) Binary Copy (4h)

(c) Parity (2h)

(d) Parity (4h)

(e) Polynomial (2h)

(f) Polynomial (4h)

**Figure 13:** Average accuracy across 4,5,6-layers models with 2 and 3-heads for Binary Copy, Parity, and Polynomial tasks.

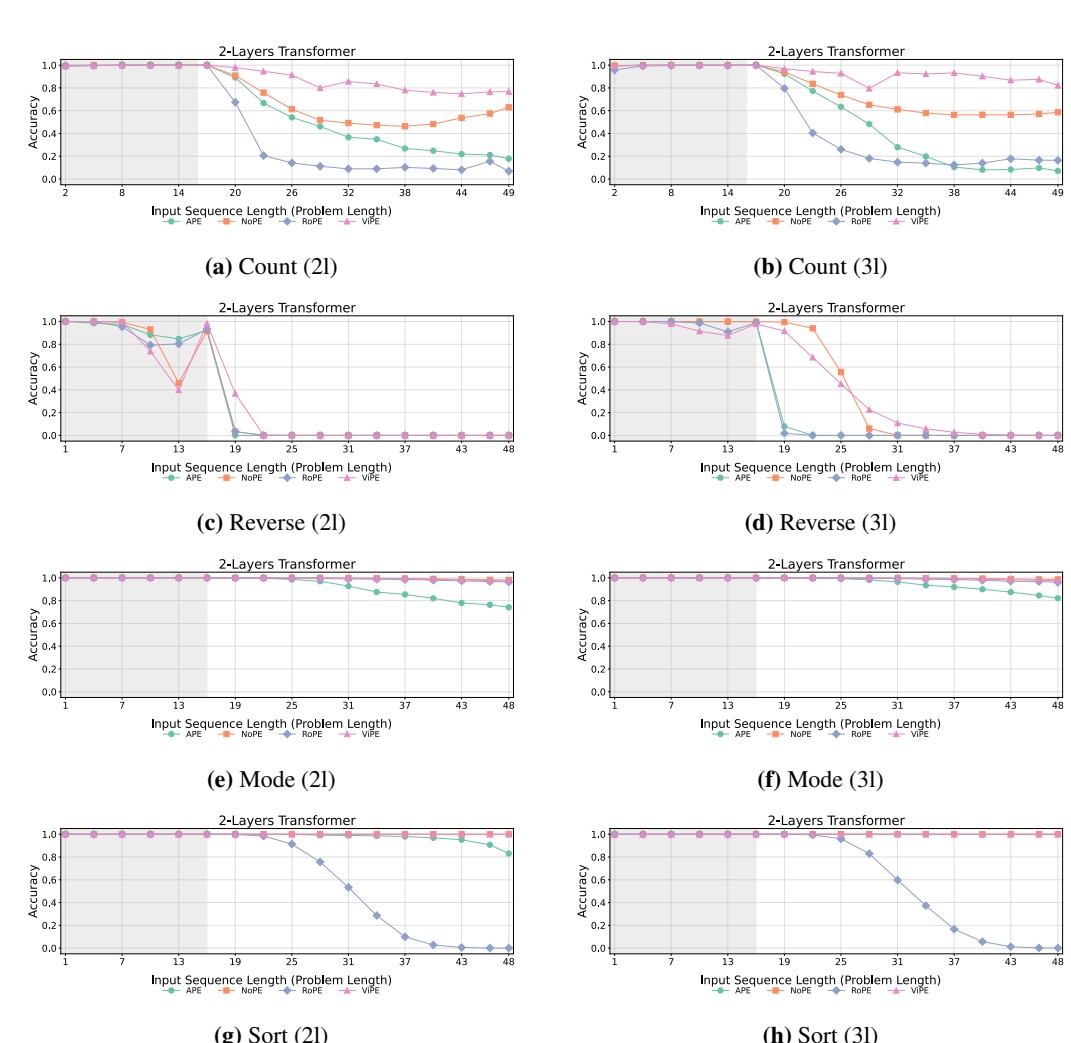

**Figure 14:** Average accuracy for Count, Reverse, Mode, and Sort tasks on 2-layer and 3-layer models with 1 head.

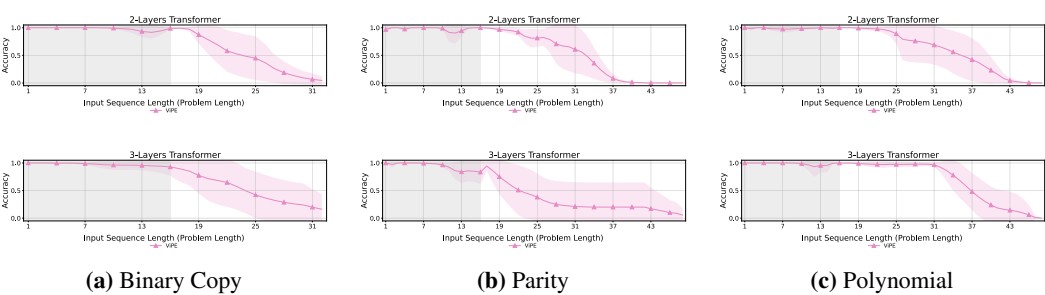

**(a)** Binary Copy          **(b)** Parity          **(c)** Polynomial

**Figure 15:** Average accuracy of ViPE in 8 different seeds for three iteratives tasks.

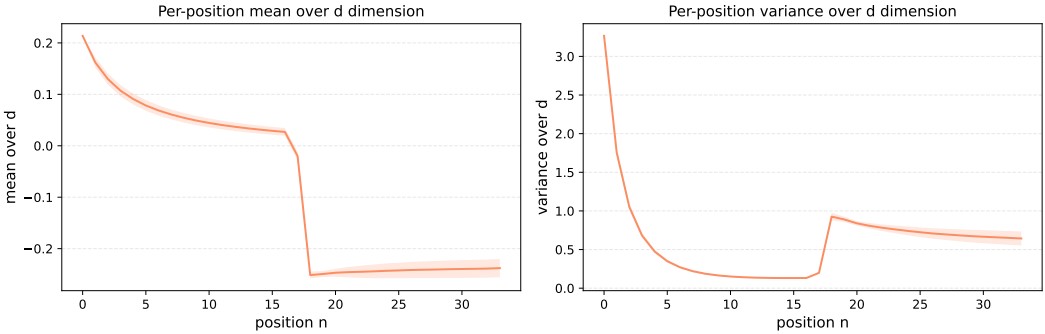

**Figure 16:** Statistics of hidden states in Polynomial Iteration with the samples of input length 16. Left: mean of $\mathbf{z_n}$. Right: variance of $\mathbf{z_n}$. Both statistics decrease gradually, consistent with the theoretical analysis. Notably, EoI acts as an anchor similar to BoS, exerting a strong influence on the statistics.

**Table 4:** Probing results for RoPE, which is worse than NoPE in the same settings. while in test, the errors increases and the coefficient metrics about abusolute position in layer 1 decreases sharply, showing that the consistency position representation breaks.

| Metric | Abs. Pos. (L1) | | Rel. Pos. (L2, EoI) | |
| --- | --- | --- | --- | --- |
| | Train | Test | Train | Test |
| RMSE | 1.9541 | 8.3372 | 2.6053 | 7.0447 |
| MAE | 1.0786 | 5.3523 | 1.8189 | 5.6159 |
| $R^2$ | 0.9434 | 0.5419 | 0.8694 | 0.6667 |
| Pearson | 0.9713 | 0.8619 | 0.9326 | 0.9046 |
| Spearman | 0.9831 | 0.8009 | 0.9360 | 0.9576 |

**Table 5:** Probing results for APE. Desipite the probe fits very well in training, the consistency position representation breaks in testing.

| Metric | Abs. Pos. (L1) | | Rel. Pos. (L2, EoI) | |
| --- | --- | --- | --- | --- |
| | Train | Test | Train | Test |
| RMSE | 0.1039 | 6.5107 | 2.3797 | 7.5786 |
| MAE | 0.0689 | 2.7649 | 1.4408 | 6.2647 |
| $R^2$ | 0.9998 | 0.7206 | 0.8910 | 0.6143 |
| Pearson | 0.9999 | 0.8791 | 0.9439 | 0.8553 |
| Spearman | 0.9985 | 0.8693 | 0.9438 | 0.8626 |

