# OpenReview forum: "Dissecting the Role of Positional Encoding in Length Generalization"
_ICLR.cc/2026/Conference — Submitted to ICLR 2026_

### Official Review · Reviewer_VJmt · 2025-10-21

**Soundness:** 3
**Presentation:** 3
**Contribution:** 2
**Rating:** 6
**Confidence:** 3

**Summary:**

This paper focuses on the impact of positional encoding strategies (ALiBi, APE, FIRE, NoPE, RoPE, T5, YARN) on length generalization of Transformers on iterative tasks (specifically Polynomial Iteration, Parity, Binary Copy). The authors propose that successful LG relies on alignment between the iterative task's computational structure and the inductive bias of the positional encoding. Their mechanistic analysis shows that many popular PEs are misaligned with iterative tasks, helping to explain why they often perform worse than NoPE. Finally, they propose modified PEs aimed at improving alignment with iterative tasks.

**Strengths:**

The mechanistic analysis is clear, convincing, and interesting. The NoPE statistical analysis nicely complements the constructive argument in Kazemnejad.

**Weaknesses:**

A limitation is studying only iterative tasks. In particular, the Logit controller and Value-side relative PE appear to be specifically designed to improve LG for iterative tasks, but their impact on LG for other types of tasks (such as the many others studied in Kazemnejad) is unclear. To be practically useful, we would hope for PEs that could improve LG on many kinds of tasks, not just a limited subset.

See also questions.

**Questions:**

Can you more explicitly position the paper relative to Kazemnejad, noting the novel contributions w.r.t. Kazemnejad? Kazemnejad et al. (2023) show the failure of LG and the relative superiority of NoPE over other PEs over a range of tasks (Fig F.5 shows (lack of) LG for Parity for NoPE, T5, ALiBi, APE). Kazemnejad further prove that NoPE can theoretically represent both absolute and relative PEs, e.g. for a specific weight configuration in the first layer, and all subsequent layers, respectively. In my reading, the novelty of the current paper lies in: a specific study of *iterative tasks* only (adding the tasks Polynomial Iteration and Binary Copy to Kazemnejad which already studies Parity), a mechanistic explanation of the specific failure-modes of various PEs for this task, and a new statistical analysis of NoPE’s ability to encode position information (distinct from Kazemnejad’s proof which relies on constructing specific weight matrices). Is this accurate?

Studying 2- and 3-layer Transformers makes sense for the mechanistic analysis where you are looking for particular expected attention patterns, but do you know whether training deeper Transformers (more layers) on the same tasks show the same behavior shown in Figure 3 (i.e. does length generalization still degrade relatively quickly OOD, with NoPE extrapolating better than other choices of PE)? The trend where the LG improves from 2- to 3-layer makes one wonder if it might continue to improve with more depth -- and whether the relative performance of the different PEs might change.

What can the study of iterative tasks tell us about other classes of tasks for which LG is desired? Can we expect the Logit controller and Value-side relative PE to improve (or at least not harm!) LG for other classes of tasks with different structure (e.g. arithmetic, etc.)?

Minor notes:
L17. positional enconding (PE) (abbrev. never introduced)
L175 Typo “Algins”

---

> ### Author Response · Authors · 2025-11-25
>
> Thank you for the insightful comment. We will address your question below.
>
> **For weakness 1**
>
> We agree with the reviewer’s summary and clarify our contributions more concisely below.
>
> **(1) Task setting and architectural choice.**
> Kazemnejad et al. [1] mainly studies Parity without scratchpad; their scratchpad version (Fig. 6) performs even worse because no-scratchpad Parity does not require position discrimination. Our work focuses on *scratchpad/CoT-style iterative tasks*, which are among the hardest settings for length generalization—tasks on which even large language models may fail.
>  To study their failure modes, we avoid hardmax attention, orthogonal embeddings, or specialized architectures. Instead, we show that a plain 2–3 layer Transformer can generalize strongly *when the PE aligns with the required computation*.
>
> **(2) Mechanistic insight.**
>  Our interpretability analysis shows that Transformers solving iterative tasks rely on a content-independent distance-computation flow in particular layers. This computation is extremely sensitive to positional misalignment: mild distortions introduced by standard PEs accumulate and cause abrupt OOD failure. This concrete failure mode had not been identified in prior work.
>
> **(3) Extending the theoretical understanding of NoPE’s positional signal.**
>  Kazemnejad et al. proves that NoPE *can represent* positional encodings under specially constructed matrices. We instead show that NoPE already induces a monotonic statistical positional signal before learning, providing a natural operational lower bound on how NoPE separates positions. Their theory gives a representational upper bound; ours gives an explanatory lower bound that helps clarify NoPE’s empirical behavior on length generalization.
>
> **(4) ViPE.**
>  ViPE was designed to validate these mechanistic observations in iterative tasks. Additional experiments show that ViPE also behaves well on other synthetic reasoning tasks and several NLP LG benchmarks, suggesting broader applicability.
>
> **In summary**, our novelty lies not only in focusing on iterative tasks, but in providing a *mechanistic explanation* for PE-induced failure modes, an analysis of NoPE’s inherent positional signal, and a *practical demonstration* that simple Transformers can generalize when correctly aligned. These insights directly motivated the design of ViPE and distinguish our work from prior studies.
>
> **For weakness 2**
>
> We trained deeper Transformers on the same tasks and evaluated their accuracy and attention patterns across PEs.
>
> Mechanistically, ViPE still induces anchor-like and ladder-like patterns in deeper models (App.D.2 Fig.10), but the computation becomes more distributed and less interpretable as depth and head count increase. Since 2–3 single-head layers already suffice for the iterative computation, additional layers likely perform auxiliary processing that is harder to trace.
>
> Empirically, across depths (App.D.2 Fig.13):
>
> - **OOD LG still degrades quickly** for APE/RoPE/NoPE.
> - **NoPE continues to extrapolate better**, but deeper models do not improve robustly; in several cases they perform worse than the minimal 2–3 layer setting.
>
> While moving from 2 to 3 layers improves NoPE (since it needs 3 layers for the required computation), further depth and multi-head attention tend to introduce more computation–task misalignment, accelerating OOD failure. This suggests that: In length generalization scenarios, model–task alignment emerges as particularly critical—rather than relying solely on model scale—which potentially supports our research. Small models may generalize well when aligned, whereas larger models—with more complex internal pathways—may suffer from severe misalignment.

---

> ### Author Response · Authors · 2025-11-25
>
> **For weakness 3**
>
> We evaluated ViPE and other PEs on larger models and NLP tasks, including SCAN and CFQ, using a GPT-2–style 6-layer, 4-head, 256-dim model. ViPE consistently outperforms APE, RoPE, and NoPE. Furthermore, following Chowdhury et al. [2], we tried to incorporate ViPE into their Mix RMonoAttn(MRMA) architecture, ViPE further improves SCAN accuracy from 0.162 to 0.293:
>
> | Dataset | APE   | RoPE  | NoPE  | ViPE      | MRMA  | MRMA+ViPE |
> | ------- | ----- | ----- | ----- | --------- | ----- | --------- |
> | CFQ     | 0.451 | 0.499 | 0.555 | **0.666** | 0.436 | 0.431     |
> | SCAN    | 0.000 | 0.021 | 0.132 | 0.150     | 0.162 | **0.293** |
>
> The table above has been added in Section 6 Table 2. We additionally include extra mathematical-task results in Appendix D.2.
>
> Deeper models introduce more opportunities for misalignment and reduced interpretability, but overall the results indicate that ViPE does not harm generalization outside iterative tasks. A likely explanation is that many reasoning and NLP tasks contain a recurrent or step-wise computational structure similar to iterative tasks, making ViPE’s alignment-oriented bias transferable.
>
> We avoid overclaiming universal benefits, but based on structural commonalities and empirical results, ViPE appears to be a potential direction for length-generalizing PEs beyond the iterative regime.
>
> **Reference**
>
> [1] Kazemnejad, et al. The impact of positional encoding on length generalization in transformers.
>
> [2] Chowdhury, et al.  Monotonic location attention for length generalization.

---

> > ### Comment · Reviewer_VJmt · 2025-11-25
> >
> > Thank you for the responses and additional results; in particular I think the new NLP experiments strengthen the paper. I will keep my positive score.

---

> > > ### Author Response · Authors · 2025-12-01
> > >
> > > We sincerely thank the reviewer for the constructive feedback and for maintaining a positive evaluation of our work. The suggestions provided have been very helpful for our future work.

---

### Official Review · Reviewer_KVcN · 2025-10-28

**Soundness:** 3
**Presentation:** 3
**Contribution:** 3
**Rating:** 6
**Confidence:** 3

**Summary:**

The paper analyzes the role of different positional encoding for length generalization on synthetic tasks like parity, binary copy, and such (typically tasks that can be accomplished by iterative local updates). The paper arguments for the misalignment between inductive biase of position encoding and the task as the key factor harming performance - and tries to propose some fixes that would created a better aligment - e.g. logit control and rescaling.

**Strengths:**

* Overall interesting analyses
* Sound lightweight extensions (ViPE) that shows effective results.

**Weaknesses:**

* Lack of benchmarking of ViPE on realistic benchamrks greatly undermines scope of the paper.
* While the paper provides some valuable insights, part of it feels somewhat "obvious" -- of course, one would think that failure to length generalize is an issue of the lacking the right inductive biase; and adding more task-specific inductive bias, or better invariance-mainetance across length increase, length generalization can be improved. This does not feel like a substantively new insight- although the key strength that redeems the paper is in proposing a potential solution.
* Similar ideas (in the context of RNNs - but the principles seem to translate) have been also explored here [1]. The benchmarks in [1] (including those from its appendix) could be have been also useful to evaluate on.
* Even the proposed method still seems to disgracefully degrades around sequence length 43-48 -- suggesting that the generalization may not scale well.

[1] Monotonic Location Attention for Length Generalization - Ray Chowdhury et al.

**Questions:**

n/a

---

> ### Author Response · Authors · 2025-11-25
>
> Thank you for the insightful comment. We will address your question below.
>
> **For weakness 1**
>
> We retrained our models on larger GPT-2–style architectures and evaluated them on both synthetic and natural language tasks. For synthetic tasks, we included *Mode, Reverse, Sort,* and *Count*. For NLP tasks, we selected SCAN and CFQ, two benchmarks commonly used to test length generalization. All experiments were conducted using a GPT-2–style model with 6 layers, 4 heads, and 256-dim embeddings.
>
> As shown below, ViPE consistently outperforms APE, RoPE, and NoPE on both NLP length-generalization datasets. Furthermore, following Chowdhury et al. [1], we tried to incorporate ViPE into their Mix RMonoAttn(MRMA) architecture, ViPE further improves SCAN accuracy from 0.162 to 0.293:
>
> | Dataset | APE   | RoPE  | NoPE  | ViPE      | MRMA  | MRMA+ViPE |
> | ------- | ----- | ----- | ----- | --------- | ----- | --------- |
> | CFQ     | 0.451 | 0.499 | 0.555 | **0.666** | 0.436 | 0.431     |
> | SCAN    | 0.000 | 0.021 | 0.132 | 0.150     | 0.162 | **0.293** |
>
> The table above has been added in Section 6 Table 2. We additionally include extra mathematical-task results in Appendix D.2.
>
> **For weakness 2**
>
> We agree that “inductive bias matters” is not a new principle. Our work relates to prior studies on positional encodings (PEs) and length generalization but approaches the problem through mechanistic interpretability within iterative tasks.
>
> - Kazemnejad et al. (2023) [2] conducted broad empirical comparisons across PEs and provided theoretical constructions showing how NoPE can learn positional distinctions. In contrast, we focus on iterative tasks whose length generalization remains challenging for modern Transformer architectures, even at realistic model scales (e.g., Zhou et al., 2023 [3]). Through mechanistic analysis, we examine how PEs shape the *internal computation flow* of Transformers, explaining *why* many PEs fail to generalize. Our theoretical view complements Kazemnejad et al.: while they show that NoPE *can learn* positional distinctions, we show that NoPE already provides an inherent, learning-free positional signal that supports alignment and generalization.
>
> - Zhou et al. (2023) analyzed the algorithmic expressivity of Transformers using RASP, identifying task limitations but not examining how different PEs affect computation. Building on mechanistic interpretability, we treat PE choice as a central computational factor.
>
> - Wu et al. (2025) [4] analyzed long-context positional biases of RoPE, ALiBi, and NoPE, but did not mainly study their consequences for length generalization.  In contrast, under real Transformer architectures, we study *how PE-induced biases interact with alignment*, providing a finer understanding of their effect on length generalization.
>
> - Our study further differs from idealized theoretical works relying on strong assumptions—such as hardmax attention or orthogonal embeddings (Huang et al., 2025 [6]; Köcher et al., 2025 [5]). we use real Transformer architectures, real PEs, and standard softmax attention, making our findings reflective of practical behavior. While prior work shows that even large models struggle on polynomial or other iterative tasks, we demonstrate that a plain 2–3 layer Transformer can generalize when the PE aligns with the computation. Attention maps and flow-level analysis reveal the computational structure and highlight alignment as the key link between theory and practice.
>
> **Reference**
>
> [1] Chowdhury, et al.  Monotonic location attention for length generalization.
>
> [2] Kazemnejad, et al. The impact of positional encoding on length generalization in transformers.
>
> [3] Zhou, et al. What algorithms can transformers learn? a study in length generalization.
>
> [4] Wu, et al. On the emergence of position bias in transformers.
>
> [5] Kocher et al. Nope: The counting power of transformers with no positional encodings.
>
> [6] Huang, et al. Transformers provably learn chain-of-thought reasoning with length generalization.

---

> > ### Author Response · Authors · 2025-11-25
> >
> > **For weakness 3**
> >
> > This is indeed a highly relevant and insightful paper. In our new experiments, we also evaluate on SCAN and CFQ, and the performance comparison is shown in the table above. The paper’s discussion on the relationship between task structure and the inductive bias introduced by different PEs shares certain conceptual similarities with our work.
> >
> > Compared with this work, our contribution focuses more on fine-grained interpretability, while that paper aims to identify an effective model architecture for length generalization. The two directions are complementary. Its model design achieves strong performance across tasks, and its perspective on PE–task interaction is valuable. We have included it in the references, and it will inform future extensions of our framework.
> >
> > **For weakness 4**
> >
> > As our analysis shows, even when a Transformer appears aligned with an iterative task, PE-induced computational bias and softmax attention can introduce noise and misalignment during length extrapolation. ViPE mitigates some issues—such as entropy growth, distance-based decay, and out-of-range positional drift—but cannot fully eliminate misalignment. In principle, if we ignore real Transformer constraints, there are many ways to achieve perfect alignment with iterative tasks: use hardmax attention and add explicit positional tags. However, such a model is useful only for theoretical analysis.
> >
> > Thus, we focus on solutions consistent with realistic PE mechanisms. Although ViPE is not perfect, it provides a practical and interpretable approach that brings Transformers closer to the alignment needed for length generalization while remaining compatible with real-world architectures.

---

> ### Comment · Reviewer_KVcN · 2025-11-26
>
> Thanks for the additional experiments.
>
> I improved my score to 8, however, the paper can still benefit from one/two experiments on fully realistic settings. Doesn't have to be anything too large scale - maybe some classical machine translation task - something as a "sanity test" that the new positional encoding at least keeps up and doesn't harm significantly - because SCAN is synthetic, and CFQ - is sort of synthetic as well although a bit more realistic. There can be some more synthetic experiments as well because they can be faster to do - like the synthetic new tasks introduced in Chowdhury, et al.  or the other tasks from Kazemnejad et al.
>
> But otherwise, the new experiments still look good and shows some promise. They mostly alleviate my primary concern in weakness 1.
>
> The responses on the other weaknesses are also appreciated. The additional discussions can benefit the paper.
>
> For future explorations, it could be also good to explore/experiment with how this may compare against newer PE-strategies like DAPE [1] and others. I know it's can be hard to consider everything, because new positional encodings come out everyday promising length extrapolation. But you may find few more salient works in the literature that have come up after Nope/Rope/Alibi that can be considered.
>
> [1] DAPE: Data-Adaptive Positional Encoding for Length Extrapolation -  Zheng et al.

---

> ### Author Response · Authors · 2025-12-01
>
> Thanks for the helpful suggestion. To further strengthen the empirical coverage, we added three additional experiments. Due to time constraints, all experiments use relatively simple configurations, but we believe their outcomes still provide insights.
>
> First, on the IWSLT14 De–En dataset, we created a length-based split by restricting the training and test sets to sequences of length 30. Due to the high computational cost, we evaluated only NoPE and ViPE. The results show that although both methods fit the training set well, their performance on the extrapolated test set remains limited, and ViPE achieves a slightly higher BLEU score than NoPE.
>
> Second, we incorporated the LEGO dataset as an additional complex synthetic reasoning benchmark. In this setting, ViPE and NoPE perform similarly, both achieving 32% accuracy, while RoPE and APE fail to generalize.
>
> Finally, we also evaluated the methods on PCFG, another complex synthetic reasoning dataset similar in nature to LEGO.
> PCFG represents a more challenging compositional reasoning setting, and ViPE achieves the best accuracy among all PEs on this task.
>
> The combined results are summarized below:
>
> | Dataset | Metric   | NoPE  | ViPE  | RoPE  | APE   |
> |---------|----------|-------|-------|-------|-------|
> | IWSLT14 | BLEU     | 0.091 | 0.118 |   –   |   –   |
> | LEGO    | Accuracy | 0.320 | 0.320 | 0.000 | 0.000 |
> | PCFG    | Accuracy | 0.200 | 0.247 | 0.232 | 0.170 |
>
> Across all experiments conducted so far, we have evaluated 12 types of tasks: mathematical reasoning tasks (polynomial iteration, parity, binary copy, mode, sort, count, reverse, LEGO, PCFG), synthetic NLP reasoning tasks (SCAN, CFQ), and a real-world MT task (IWSLT14). In the vast majority of these settings, ViPE matches or outperforms other PEs in terms of length generalization, reinforcing our position that ViPE at least does not harm model performance.
>
>
> For future research, we plan to continue investigating positional encodings from an interpretability perspective, and to incorporate more recent PE methods for a more comprehensive comparative study.
>
> We sincerely thank the reviewer for the constructive feedback and for taking the time to re-evaluate our submission. We greatly appreciate the score increase and the insightful suggestions, which have helped strengthen the paper.

---

### Official Review · Reviewer_ivHQ · 2025-11-01

**Soundness:** 2
**Presentation:** 2
**Contribution:** 2
**Rating:** 4
**Confidence:** 2

**Summary:**

This paper investigates the mechanisms behind length generalization in Transformers, proposing that LG depends on the alignment between a model’s inductive bias and the computational structure of the task. Through synthetic experiments on iterative reasoning tasks, the authors analyze various positional encodings, including RoPE,  NoPE, and find that most fail to generalize to longer sequences. They show that NoPE can partially achieve LG via implicit positional signals emerging from hidden-state statistics and contextual token distributions, though these signals decay with length. Building on this analysis, the paper introduces ViPE combining value-side relative coding and logit rescaling, aligning model bias with task structure and substantially improving extrapolation performance.

**Strengths:**

1. The paper provides a novel explanation of how NoPE implicitly encodes positional information through hidden-state statistics and contextual token distributions, contributing theoretical clarity to understanding Transformers without explicit positional encodings. The proposed method ViPE introduces value-side relative encoding and logit rescaling, significantly improving length extrapolation and demonstrating the practical value of aligning model inductive bias with task structure.

2. The experiments are thorough and clearly presented, covering multiple positional encodings and iterative tasks. The visualization of attention maps and performance degradation effectively supports the paper’s main claims about misalignment and fragility in length extrapolation.

3. The paper offers a fresh view by framing length generalization as an alignment problem between a model’s inductive bias and the computational structure of the task, providing an insightful analytical framework.

**Weaknesses:**

1. All experiments are conducted solely on synthetic iterative tasks, leaving it unclear whether the conclusions generalize to natural language or more complex reasoning tasks. This considerably limits the paper’s practical value. For instance, in general length generalization settings using pretrained models (e.g., Qwen2.5), would the attention maps still exhibit such clear structural patterns?

2. Since the paper focuses exclusively on synthetic tasks, and ViPE appears somewhat tailored to tasks with precise computational structures (is that correct?), I wonder how the authors envision its performance on more typical tasks, including natural language and general length generalization benchmarks. Given resource and time constraints, additional experiments are unnecessary, but I would appreciate the authors’ perspective on this point.

3. The analysis of NoPE seems to show only that NoPE can use statistical information to distinguish positions, but not that it actually does so in practice. The experiments in Section 5.3 merely demonstrate that NoPE encodes absolute and relative positions. Should this be considered only a lower bound on NoPE’s capability? That said, the authors’ analysis is valuable in that it inspired the design of ViPE, which is a positive contribution.

However, I’m not fully confident in my own judgment, and I'm willing to adjust my score after seeing other reviewers’ comments and the authors’ rebuttal.

**Questions:**

See weakness.

---

> ### Author Response · Authors · 2025-11-25
>
> Thank you for the insightful comment. We will address your question below.
>
> **For weakness 1**
>
> We evaluated ViPE and other PEs on larger models and NLP tasks, including SCAN and CFQ, using a GPT-2–style 6-layer, 4-head, 256-dim model. ViPE consistently outperforms APE, RoPE, and NoPE. Furthermore, following Chowdhury et al. [1], we tried to incorporate ViPE into their Mix RMonoAttn(MRMA) architecture, ViPE further improves SCAN accuracy from 0.162 to 0.293:
>
> | Dataset | APE   | RoPE  | NoPE  | ViPE      | MRMA  | MRMA+ViPE |
> | ------- | ----- | ----- | ----- | --------- | ----- | --------- |
> | CFQ     | 0.451 | 0.499 | 0.555 | **0.666** | 0.436 | 0.431     |
> | SCAN    | 0.000 | 0.021 | 0.132 | 0.150     | 0.162 | **0.293** |
>
> The table above has been added in Section 6 Table 2. We additionally include extra mathematical-task results in Appendix D.2.
>
> We examined attention maps of the 6-layer GPT-2 model.
>
> - On iterative tasks (Fig.10), some heads still display the *anchor-based* or *ladder-like* structures described in the paper, though naturally less clean than in the 2-layer setting.
> - On SCAN and CFQ (Fig.11), the computation is more complex, so the patterns are not identical to those in the main paper; however, the model still focuses on specific tokens rather than exhibiting unstructured attention.
>
> Importantly, larger model size does not guarantee better length extrapolation. As shown in App.D.2 Fig.13, making the network deeper or wider may not improve performance on iterative tasks and can even degrade extrapolation in some cases. Consistent with this trend, we also experimented with GPT-2 Large, but after training, the model exhibited no meaningful length generalization on SCAN for any PE method we tested, including ViPE.
>
> This suggests that: In length generalization scenarios, model–task alignment emerges as particularly critical—rather than relying solely on model scale—which potentially supports our research. Small models may generalize well when aligned, whereas larger models—with more complex internal pathways—may suffer from severe misalignment.
>
> **For weakness 2**
>
> As shown earlier, ViPE does demonstrate promising behavior on NLP benchmarks. However, for deep Transformer architectures and complex NLP tasks, the underlying task structure and inductive biases are much harder to characterize. Therefore, we do not wish to overclaim that ViPE will consistently outperform other PEs on all natural language tasks.
>
> Although natural language tasks appear more complex, many large models that perform well on such tasks still fail to extrapolate on seemingly simple iterative tasks. This contrast indicates that different tasks impose fundamentally different structural requirements, and therefore benefit from different inductive biases. For example, many language tasks rely primarily on local context, where long-range interactions introduce more noise than signal; in such settings, distance-decay encodings such as RoPE may align better with the underlying computation. Conversely, long-context or retrieval-style reasoning requires stable cross-token connectivity, where ViPE’s non-decaying behavior *may* offer advantages.
>
> From the perspective of our study, ViPE was intentionally designed to test how correct model–task alignment supports length generalization in iterative tasks. However, many NLP reasoning tasks can be decomposed into a recurrent or iterative computational structure that resembles the alignment demands of our synthetic tasks. For this reason, while we do not expect ViPE to perform well on *all* natural language tasks, we believe it holds meaningful potential for NLP tasks whose internal computation is structurally similar to iterative reasoning.
>
> **For weakness 3**
>
> Our theoretical result should indeed be interpreted as a lower bound on NoPE’s positional capability: even under the most primitive assumptions (e.g., uniform attention), NoPE already introduces weak but monotonic positional distinctions. This does not aim to fully characterize NoPE’s learned representations, but rather to isolate its inherent positional signal.
>
> Beyond Section 5.3. Similarly, Fig.16 in Appendix D.2 shows that the mean and variance of NoPE hidden states (after BoS and EoI) follow consistent monotonic trajectories as sequence length grows. These empirical patterns provide further evidence that NoPE’s inherent positional distinctions are preserved in real Transformer settings.
>
> Taken together, these results suggest that NoPE naturally forms a weak but robust positional signal, which helps explain its empirical length-generalization behavior. And indeed inspired the design of ViPE.
>
> **Reference**
>
> [1] Chowdhury et al.  Monotonic location attention for length generalization.

---

### Official Review · Reviewer_nP5W · 2025-11-02

**Soundness:** 3
**Presentation:** 2
**Contribution:** 2
**Rating:** 2
**Confidence:** 4

**Summary:**

The paper proposes a formal bridge between LLMs and Algorithmic Information Theory (AIT), arguing that (i) LLM training ``computationally approaches'' the Solomonoff prior by interpreting loss minimization as an implicit program-length optimization, and (ii) LLM next-token prediction acts as a computable surrogate for Solomonoff induction under an approximation assumption. It then uses this lens to give unified explanations for in-context learning, few-shot learning, and scaling laws, and introduces a few-shot example selection rule (pick low-confidence correct examples). This selection rule improves accuracy on three text-classification benchmarks (SMS, emotion, and AG News) with Qwen and Llama models.

**Strengths:**

1. The paper presents an interesting theoretical connection between LLMs and the Solomonoff prior.

2. The paper argues how context, few examples, and more compute/parameters drive predictions toward a target distribution. The authors show that low-confidence correct examples are more beneficial for ICL than easy examples.

**Weaknesses:**

1. The novelty of the paper lies in the connection between LLMs and Solomonoff induction, but from the perspective of the evaluation (ICL with correct samples that have low confidence) this has limited novelty.

2. Experiments evaluate only few-shot classification with confidence-based selection across three datasets and four instruction-tuned models. There’s no ablation on prompt length, budget K, alternative selection criteria (entropy, gradient-free influence, diversity), or tasks beyond classification.

3. Results show consistent gains for ``low-confidence'' selection, but it is not clear where the models make mistakes. An error analysis would be good.

4. The paper would benefit from a proofread. See for example, ``Thus, the LLM as a whole functions as a deterministic Turing machine.''

**Questions:**

What kinds of errors do the models make?

How are the samples in the prompt selected (beyond being correct and low confidence?)

How many samples in the prompt are there?

---

> ### Author Response · Authors · 2025-11-25
>
> Thank you for the insightful comment. We will address your question below.
>
> **How would the approach perform on more realistic tasks (e.g., NLP)?**
>
> We evaluated ViPE on larger models and NLP tasks, including SCAN and CFQ, using a GPT-2–style 6-layer, 4-head, 256-dim model. ViPE consistently outperforms APE, RoPE, and NoPE. Furthermore, following Chowdhury et al. [1], we tried to incorporate ViPE into their Mix RMonoAttn (MRMA) architecture, ViPE further improves SCAN accuracy from 0.162 to 0.293:
>
> | Dataset | APE   | RoPE  | NoPE  | ViPE      | MRMA  | MRMA+ViPE |
> | ------- | ----- | ----- | ----- | --------- | ----- | --------- |
> | CFQ     | 0.451 | 0.499 | 0.555 | **0.666** | 0.436 | 0.431     |
> | SCAN    | 0.000 | 0.021 | 0.132 | 0.150     | 0.162 | **0.293** |
>
> This table is now included in Section 6 (Table 2). **Additional mathematical-task results appear in Appendix D.2.**
>
> While these gains suggest practical potential, we avoid attributing them solely to alignment, since deeper multi-head Transformers are not fully interpretable. ViPE aims to test whether alignment-based improvements are real and indicate a direction for length generalization. Nevertheless, it is useful for future work.
>
> **How does the approach connect to other PE-based modeling?**
>
> Our work relates to prior studies on PEs and length generalization but analyzes the problem through mechanistic interpretability within iterative tasks.
>
> - Kazemnejad et al. (2023) [2] conducted broad comparisons across PEs and showed how NoPE can learn positional distinctions. In contrast, we study iterative tasks whose length generalization remains challenging for modern Transformers, even at realistic scales (e.g., Zhou et al., 2023 [3]). We analyze how PEs shape the internal computation flow, explaining why many PEs fail. Our theoretical view complements Kazemnejad et al.: they show NoPE can learn positional distinctions; we show NoPE already provides an inherent, learning-free positional signal supporting alignment and generalization.
>
> - Zhou et al. (2023) analyzed the expressivity of Transformers using RASP but did not analyze PE effects. Building on mechanistic interpretability, we treat PE choice as a central computational factor.
>
> - Wu et al. (2025) [4] analyzed long-context positional biases of RoPE, ALiBi, and NoPE, without focusing on length generalization. We instead study how PE-induced biases interact with alignment in real Transformer architectures, providing a finer understanding of length generalization.
>
> - Our study further differs from idealized theoretical works relying on strong assumptions—such as hardmax attention or orthogonal embeddings (Huang et al., 2025 [6]; Köcher et al., 2025 [5]). we use real Transformer architectures, real PEs, and standard softmax attention, making our findings reflective of practical behavior. While prior work shows that even large models struggle on polynomial or other iterative tasks, we demonstrate that a plain 2–3 layer Transformer can generalize when the PE aligns with the computation. Attention maps and flow-level analysis reveal the computational structure and highlight alignment as the key link between theory and practice.
>
> **How sensitive are the results to the assumption of uniform attention?**
>
> We agree that assuming uniform attention is a strong simplification used mainly to facilitate theoretical analysis. In our view, the theoretical result should be interpreted as a lower bound on the positional information that NoPE inherently provides: even in its most primitive form (with uniform attention), NoPE already carries distinguishable positional signals. This complements Kazemnejad et al. (2023).
>
> While the theory cannot assume the full expressiveness of real attention mechanisms, we conducted several experiments to verify that the theoretical intuition aligns with empirical behavior.
>
> Specifically, in Section 5.3, a two-layer Transformer allows a linear probe to recover linearly separable, monotonic positional structure, and this trend persists under extrapolated sequence lengths. Similarly, Figure 16 in Appendix D.2 shows that the mean and variance of NoPE hidden states for tokens after BoS and EoI follow a consistent monotonic trajectory. This empirical trend can be viewed as an extension of the “uniform-attention positional effect” into real Transformer settings, reinforcing the consistency between our theoretical analysis and actual model behavior.
>
> **Reference**
>
> [1] Chowdhury, et al.  Monotonic location attention for length generalization.
>
> [2] Kazemnejad, et al. The impact of positional encoding on length generalization in transformers.
>
> [3] Zhou, et al. What algorithms can transformers learn? a study in length generalization.
>
> [4] Wu, et al. On the emergence of position bias in transformers.
>
> [5] Kocher et al. Nope: The counting power of transformers with no positional encodings.
>
> [6] Huang, et al. Transformers provably learn chain-of-thought reasoning with length generalization.

---

> > ### Author Response · Authors · 2025-11-27
> >
> > To address the concern that ViPE is only evaluated on the polynomial iteration task, Appendix D.2 (Figs. 12, 13 and 14) reports results on seven mathematical reasoning tasks — polynomial iteration, parity, binary copy, mode, count, reverse, and sort — in addition to the NLP length-generalization benchmarks SCAN and CFQ discussed above. Across these nine tasks, ViPE matches or outperforms APE, RoPE, and NoPE in most settings.
> >
> > Although ViPE is designed specifically to improve alignment for iterative tasks (polynomial, parity, and binary copy), these experiments indicate that it does not harm, and in several cases improves, generalization on the other mathematical reasoning and NLP tasks. A plausible explanation is that many of these tasks can be solved via step-wise computations similar to those in our iterative setting, making ViPE’s alignment-oriented inductive bias transferable beyond the synthetic algorithmic benchmarks we analyze theoretically.

---

> ### Author Response · Authors · 2025-11-27
>
> Dear Reviewer  nP5W,
>
> Thank you again for your thoughtful comments on our submission. As the discussion period is coming to a close, we would like to ask whether our responses have addressed your concerns. If any points remain unclear or if you have additional questions, we would greatly appreciate your guidance and will be happy to clarify further. If you feel that our revisions and explanations have resolved the issues you raised, we would be grateful if you could consider updating your evaluation accordingly.
>
> We sincerely look forward to your feedback, and thank you for your time and effort throughout this process.
>
> Best regards,
>
> Authors of Submission17734

---

### Author Response · Authors · 2025-12-01
**Summary of the rebuttal**

Dear AC,

Thank you for taking the time to review our submission. Below is a summary of our rebuttal and the main points addressed.

We received comments from four reviewers, and we found that most concerns fall into the following categories:

1. **Need for broader experiments.**
    Reviewers requested additional evidence beyond the simple Transformer settings and specific iterative tasks used in our interpretability analysis. In particular, they asked whether the patterns observed in small models carry over to larger architectures, and whether ViPE behaves well on more complex reasoning tasks or realistic NLP tasks.
2. **Clarifying novelty relative to prior work.**
    Several reviewers asked us to more clearly articulate how our contributions differ from and complement existing studies on positional encodings and length generalization.
3. **Scope of the theoretical analysis for NoPE.**
    Some reviewers raised questions about whether the theoretical lower-bound characterization of NoPE under strong assumptions meaningfully reflects its behavior in practical settings.

### Our responses to these concerns are summarized as follows:

1. **Expanded experiments.**
    We conducted substantially broader evaluations across models and tasks.
    Using GPT-2–style models up to 6 layers and 4 heads, we tested 12 datasets:

   - Mathematical reasoning tasks: *polynomial iteration, parity, binary copy, mode, sort, count, reverse, LEGO, PCFG*
   - Synthetic NLP reasoning tasks: *SCAN*, *CFQ*
   - A real-world MT task: *IWSLT14*

Across the vast majority of these settings, ViPE matches or outperforms other PEs in length generalization. We also visualized attention maps for several tasks. While deeper multi-head models are less interpretable, certain heads still exhibit strong attention to key tokens, suggesting continuity between small-model patterns and larger models.

2. **Clarifying our novelty.**
    We reiterate that although positional encoding research is extensive, our contributions lie in the *mechanistic* understanding of PE-induced misalignment, our explanation of NoPE’s inherent positional signal, and the demonstration that simple Transformers can generalize strongly when aligned with the underlying computation. These insights complement rather than duplicate prior work.

3. **Clarifying the theoretical scope of NoPE.**
    We noted that our analysis provides a *lower bound* on NoPE’s positional discrimination ability. This complements Kazemnejad et al.’s representational analysis.
    Although theory alone cannot guarantee the emergence of these properties during learning, our Section 5.2 experiments demonstrate that the predicted monotonic statistical signal does appear in practice, suggesting transferable value.

### Feedback from reviewers

Two reviewers (KVcN and VJmt) engaged with our rebuttal before the OpenReview leak issue.

- **VJmt** stated that our responses addressed his concerns and that he would maintain a *positive score*.
- **KVcN** expressed that the new experiments were helpful, improved our score to **8**, and encouraged further extensions.

Following KVcN’s additional suggestions, we conducted three more experiments (IWSLT14, LEGO and PCFG), which further support the conclusion that ViPE at least does not harm model performance.

We hope this summary is helpful for your assessment. Thank you again for your time and effort.

Best regards,

Authors of Submission17734

---

### Meta-Review · Area_Chair_pjg5 · 2026-01-06

**Summary:**

## Summary
This paper examines length generalization (LG) of Transformers by testing the alignment between task structure and model inductive bias (primarily focusing on different choices of positional encoding). For iterative tasks, the paper analyzes whether different positional encodings (PEs) yield the attention patterns required by the iterative task, and concludes that misalignment happens when we extrapolate the length, and it arises from the biases of PEs and softmax attention. The paper further analyzes NoPE, which shows better LG performance, and studies how NoPE uses context statistics to help encode position information. Lastly, the authors propose a new PE called ViPE, which uses value-side relative coding with logit rescaling. The new PE is tested on the synthetic iterative task as well as real datasets such as SCAN and CFQ.

## Reviewer Concerns
Major concerns raised by reviewers can be summarized as follows.
- **Limited evaluation of ViPE**. All reviewers pointed out the empirical evaluation was solely done on the synthetic iterative tasks. They asked if ViPE can be used for other synthetic (e.g., arithmetic) algorithmic tasks that require length generalization, or if there can be any other realistic benchmarks for which ViPE can be evaluated.
- **Novelty/significance of NoPE analysis**. Reviewer nP5W pointed out that the assumption of uniform attention is not realistic, Reviewer ivHQ commented that the connection of theory and experiments is unclear, and Reviewer VJmt asked clarifying questions on the novelty of this paper relative to Kazemnejad et al. (2023).
- **Does ViPE solve the identified problems?** Reviewer nP5W asked whether ViPE remedies the two identified problems: i) structural bias from softmax attention and (ii) computational bias from PEs, and if the authors can evaluate this aspect comprehensively. While Reviewer ivHQ commented that the NoPE analysis inspired the design of ViPE, from my own reading, this was not very obvious from the main text.
- **New insights**. Reviewer KVcN pointed out that part of the observations in this paper feels somewhat obvious.

**Reviewer Concerns:**

Unfortunately, the discussion period closed before we received feedback from two out of the four reviewers. Based on my reading, the authors’ responses can be summarized as follows.
- **Limited evaluation of ViPE**. The authors evaluated ViPE on larger models and NLP tasks, including SCAN and CFQ. The authors also added several mathematical task results in Appendix D.2. ViPE shows good performance compared to APE, RoPE and NoPE in both CFQ and SCAN benchmarks, but has mixed results when adding MRMA to ViPE. Reviewer KVcN commented that it would be good to add other NLP tasks to demonstrate that the new positional encoding at least keeps up and doesn't harm significantly. The authors responded with additional experiments on IWSLT14, LEGO, PCFG.

- **Novelty/significance of NoPE analysis**.
  - For Reviewers nP5W and ivHQ’s questions, the authors claimed that the theoretical result should be interpreted as a lower bound on the positional information that NoPE inherently provides. I am **not** fully convinced by this response, and it is unclear to me what kind of “lower bound” the authors are referring to. For different attention weights, the averages of value vectors can have different statistical properties, and may fail to encode the position.
  - For Reviewer VJmt’s question, the authors mostly agreed with the reviewer’s summary but clarified some other aspects such as ViPE.

- **Does ViPE solve the identified problems?** The authors **did not respond** to Reviewer nP5W’s question of whether the structural bias and computational bias of ViPE can be evaluated. Even after the rebuttal, I feel that the connection to ViPE’s design and correction of the structural/computational bias is slightly unclear. I understand that introducing $s$ and $\lambda_{ni}$ could alleviate the biases, but what about other new elements in the design? Does ViPE work as intended?

- **New insights**. The authors clarified the difference between the approaches taken in this paper and existing results: LG through the lens of mechanistic interpretability.

**Reviewer Scores:**

The initial reviews had scores 6/6/4/2. Reviewers VJmt and KVcN (both with initial score 6) responded to the rebuttal and Reviewer KVcN raised the  score to 8 after the discussion. Here is what I expect about the reviewers’ final assessment, focusing on the negative reviewers:
- I believe some concerns raised by **Reviewer nP5W** (score 2) remain even after the rebuttal. The concern on “Does ViPE solve the identified problems?” was not answered by the authors, and to me, the explanation that uniform attention weights serve as a lower bound is not fully convincing either.
- **Reviewer ivHQ**’s (score 4) concerns were better addressed, especially by supplementing the paper with additional experimental results. The reviewer may have increased the score to 6.

After my reading of the paper, if I were the reviewer, my final evaluation would have been a score of 4.
1. While I believe that the mechanistic interpretability viewpoint in length generalization is new in the literature, I’m not fully convinced if the suggested attention pattern is “the only” way to solve the iterative tasks. It appears to be a natural pathway to take, but do we know if there is really no alternative path at all? Given this, the connection between the breakdown of attention patterns and the deterioration of LG performance is less formal—although the connection is intuitive.

2. The analysis in Section 5 looks somewhat straightforward to me and does not add much insight. The authors frame the propositions as “lower bounds” but I am not particularly convinced why this framing is appropriate (see above). It is mentioned in the beginning of Section 5.1 that “Kazemnejad et al. (2023) showed that the first layer of NoPE captures absolute positions, while the second layer captures relative positions, …” Then, what is the contribution of Section 5.3? The connection of Section 5.3 and the earlier theory part is also weak, and the inspiration from NoPE analysis in the design of ViPE should be better explained.

3. The authors put extensive efforts into running additional experiments and they substantially improved the empirical support for ViPE. However, if we were to accept this paper mainly due to the new position encoding method ViPE, the method must go through greater scrutiny and comparative experimental analysis. As noted by Reviewers nP5W and KVcN, the paper does not compare the paper against other newer position encoding methods. For NLP experiments, the methods should be compared against stronger benchmarks such as FIRE and DAPE. For synthetic tasks, it would be great to compare against Position Coupling and Abacus on arithmetic tasks. While the additional experiments make ViPE a more promising approach, more extensive comparisons need to be carried out if the paper were to be accepted based on the invention of ViPE.

Overall, the paper makes nontrivial contributions but sits slightly below the acceptance threshold. I cannot recommend acceptance at this time.

---

### Decision · Program_Chairs · 2026-01-26

Reject